

# Evaluation of Unified Model Rainfall Forecasts over the Western Ghats and North East states of India

Kuldeep Sharma[1], Sushant Kumar[1], Raghavendra Ashrit[1], Sean Milton[2], Ashis K. Mitra[1] and Ekkattil N. Rajagopal[1]

[1]National Centre for Medium Range Weather Forecasting, A-50, Sector-62, Noida 201309
[2]Met Office, FitzRoy Road, Exeter Devon, EX1 3PB, United Kingdom

*Correspondence to:* Kuldeep Sharma (kuldeep@ncmrwf.gov.in)

**Abstract**

Prediction of heavy rains associated with orography is still a challenge, even for the most advanced state-of-art high-resolution Numerical Weather Prediction (NWP) modeling systems. The aim of this study is to evaluate the performance of UK Met Office **U**nified **M**odel (**UM**) in predicting heavy and very heavy rainfall exceeding $80^{th}$ and $90^{th}$ percentiles which occurs mainly due to the forced ascent of air parcels over the mountainous regions of the Western Ghats (WGs) and North East (NE) – states of India during the monsoon seasons of 2007 to 2018. Apart from the major upgrades in the dynamical core of UM from New Dynamics (ND) to Even Newer Dynamics for General Atmospheric Modeling of the environment (ENDGame), the horizontal resolution of the model has been increased from 40 km and 50 vertical levels in 2007 to 10km and 70 vertical levels in 2018. In general, it is expected that the prediction of heavy rainfall events improves with increased horizontal resolution of the model. The evaluation based on verification metrics, including Probability of Detection (POD), False Alarm Ratio (FAR), Frequency Bias (Bias) and Critical Success Index (CSI), indicate that model rainfall forecasts from 2007 to 2018  have improved from 0.29 to 0.38 (CSI), 0.45 to 0.55 (POD) and 0.55 to 0.45 in the case of FAR over WGs for rainfall exceeding the $80^{th}$ percentile (CAT-1) in the Day-1 forecast. Additionally, the Symmetric Extremal Dependence Index (SEDI) is also used with special emphasis on verification of extreme and rare events. SEDI also shows an improvement





from 0.47 to 0.62 and 0.16 to 0.41 over WGs and NE-states during the period of study,
suggesting an improved skill of predicting heavy rains over the mountains. It has also been found
that the improvement is consistent and comparatively higher over WGs than NE-states.
**Key Words:** Orographic rain, Unified Model, Categorical verification, extreme rain, rainfall
forecast, NWP
**1.  Introduction**
Orography is the primary cause of up-lift of air parcels together with convectively driven rainfall
in mountainous regions (Flynn et al., 2017). The spells of heavy orographic rainfall may induce
landslides and flash flooding which lead to tremendous damage to the lives, property,
infrastructure, environment and local economy. One of the most tragic landslides occurred in
Kedarnath (Uttarkhand, India) in 2013 which led to more than 1000 deaths and 61000 stranded
(Dube et al., 2014). During the last decade, the number of landslide incidences over India has
increased and it contributes 16% of all rainfall-triggered landslides in the global dataset (Froude and
Petley, 2018). The accurate prediction of this heavy rainfall with enough lead time over mountains
can help in mitigation and precautions towards rainfall induced disasters.
Forecasting of this orographically induced heavy rainfall is one of the most challenging problems
for numerical weather prediction (NWP) models. This is because of the complexity of the
meteorological phenomena occurring over the orographic regions and the difficulty of obtaining
detailed and precise observational data sets. (Smith et al., 1997, Mecklenburg et al., 2000, Lin et.
al, 2001). This leads to the poor representation initial conditions required to run the NWP model
(Panziera et al., 2011). However, there is a significant improvement in the forecasting skill of
NWP models in recent times. Some of these improvements can be attributed to the increased
horizontal and vertical resolutions as well as improved physics parameterization schemes



(Sharma et. al. 2017), while major credit to the substantial improvements in weather forecasting
goes to the sophisticated data assimilation systems which utilize the satellite data.
The Indian subcontinent is highly vulnerable to heavy rainfall events. Most of the heavy and
extreme rainfall events occur during the southwest monsoon season (June to September, JJAS).
Western Ghats (WGs), North-Eastern (NE) states (Assam, Meghalaya, Mizoram, Arunachal
Pradesh, Sikkim, Manipur and Tripura) of India and central India are the most prominent regions
of heavy rainfall (Pattanaik and Rajeevan 2010). Central India receives rainfall mainly due to the
Low Pressure Systems (LPS) and Monsoon Depressions (MD) that form over the Bay of Bengal
(BoB) and move towards the west north-westward during JJAS (Goswami et al 2006; Sikka,
2006; Ajaymohan et al., 2010; Krishnamurthy and Ajaymohan, 2010) and only on very few
occasions do these LPS and MD's move northwards to produce a significant amount of rainfall
over the NE states. The WGs and the NE states of India are regions characterized by steep
orography and the heavy rainfall in these regions are often due to forced ascent of air parcels
over the mountains. These two mountainous regions of India have the highest annual rainfall
(Rao, 1976, Parthasarathy et al., 1995). The WGs are aligned north-south along the western coast
of India extending from Gujarat to Kerala with a narrow zonal width and steep rising western
face with the highest peak (2.6 Km) named Anamudi and located in Kerala. The north-east
region is dominated by the Eastern Himalayan mountain range. Geographically, two-thirds of the
area is hilly terrain interspersed with valleys and plains. The mean summer monsoon rainfall
over NE-States is ~151.3 cm which is much larger than the all India average (86.5cm)
(Parthasarathy et al., 1995) making it a potential zone for hydropower.
The WGs plays a dominant role in modulating the southwest monsoon, which in turn modulates
the regional climate (Gunnell 1997), as its first encounter on landfall over India is with these



mountain chains. Evaluation of the operational Unified Model (UM) rainfall forecast over India,
using multiple monsoon seasons, is documented in two recent studies. Kuldeep et al, (2017)
report improved skill of predicting heavy rainfall (>2 and >5 cm/day ) over India (Core Monsoon
Zone: 18–28N, 68–88E). In another study, Kuldeep et al (2019) document the spatial verification
of rainfall using Contiguous Rain Areas (CRA) method over different regions of India. Here,
evaluation of operational UM rainfall forecasts is focused on mountainous regions of India (over
WGs and NE-states). The study period extends over twelve monsoon seasons (2007-2018).
Evaluation is carried out with special emphasis on heavy rainfall. Unlike earlier studies, the
verification is based on quintile based rainfall thresholds rather than absolute rainfall amounts.
During 2007-18, there has been considerable interannual variability in the monsoon. India
Meteorological Department (IMD) reports show that during 2007,2008,2010 and 2011-13,
monsoon rainfall was  above normal,  while it was below normal during 2009 and 2014-18.  The
rainfall events exceeding two thresholds, the $80^{th}$ (hereafter CAT-1) and $90^{th}$ percentiles
(hereafter CAT-2) have been chosen to verify the forecast produced by the UM. For verification
based on percentiles the fraction of events classified as 'yes' are identical for different locations
or times of the year (Hamill and Juras, 2006), regardless of whether the climatological means
and variances are large or small. The rationale for choosing these rainfall thresholds of CAT-1
and CAT-2 based on percentiles is discussed in section 4.
**2. Data and Methodology**
**2.1 Observed Data**
The availability of daily rainfall data for long climatological periods is crucial for understanding
the components and processes related to the Indian monsoon. Daily rainfall associated with
orography, low-pressure systems and monsoon depressions contribute significantly to the total



seasonal rainfall. Orography plays a crucial role; the validation of numerical models requires
accurate rainfall information over land and adjoining seas (Mitra et al., 2013). The major data
sources of the rainfall are rain gauge, radar and satellite estimates (Ebert et al., 2003; Mitra et al.,
2009). Although the rain gauge network is not evenly spread in space and often very sparse over
unpopulated regions, particularly in mountainous areas, rainfall measurements from rain gauges
remain the most reliable data sources over land as they have good time resolution and provide an
accurate estimate of ground truth at a particular location. The improved representation of heavy
rainfall events due to an enhanced rain gauge network over WGs and NE-states have been
recently reported in Pai et al. (2014).
The period of the observed dataset used in this study is the monsoon season (JJAS) from 2007 to
2018. The two domains selected for the study are WGs (72-78°E, 8-23°N) and NE- states (88-
100°E, 21-30°N). The verification has been carried out over Indian land points only.
The gridded daily rainfall data set obtained from IMD for the period 2007–2011 is used in the
present study. The geographical distribution of IMD's rain gauges on any typical day over India
during the monsoon is shown in Figure 1(a). The boxes (WGs and NE-states) represent the
domains chosen for this study. The zoomed plots of WGs and NE-states are also displayed in
Figure 1(b) and (c). The number of grid points (land only) over WGs and NE-states used in the
present study are 475 and 403 respectively.  The number of rain gauge stations on any typical
day over WG and NE-states are 796 and 132 respectively. The Shepard interpolation technique
(1968), also discussed in Rajeevan et al. (2006), has been adopted for the gridding this rainfall
data. During the monsoon seasons of 2012-2018, NCMRWF-IMD (National Centre for Medium
Range Weather Forecasting - Indian Meteorological Department) merged satellite-rainfall
analyses have been used. For the monsoon seasons of 2012-15, NCMRWF-IMD rainfall data are





the merged product of near-real-time Tropical Rainfall Measuring Mission Multi-satellite
Precipitation Analysis (TMPA)-3B42 and rain gauge data from the India Meteorological
Department (IMD) using an objective analysis scheme (NMSG; Mitra et al. 2009). For the period
2016-2018, the rainfall estimates from Global Precipitation Measurement (GPM) satellite have
been used to merge with IMD's rain gauge stations to characterize the best rainfall estimates
over the Indian region. The spatial resolution of the data is at 0.5° x 0.5°. However, the spatial
resolution of rainfall data from the monsoon season of 2016 onwards is available originally at a
horizontal resolution of 0.25°, but we have interpolated this data set using a bilinear interpolation
technique at a spatial resolution of 0.5° to make a uniform rainfall data series throughout the
study. This merged data set represents the Indian monsoon rainfall more realistically and is
superior to other available rainfall data sets over the Indian monsoon region because it uses
additional local rain gauge observations (Mitra et al. 2013), and consequently provides a better
baseline for NWP model validation and monsoon model development.
**2.2 Description of the NWP Modelling System and Forecast Dataset**
The Unified Model at the UK Met Office is the numerical modeling system developed for the
seamless prediction of weather and climate systems (Davies et al. 2005; Brown et al. 2012,
Wood et al. 2014; Met Office 2014). This 'seamless' prediction system implies that the same
model with slightly different configurations (e.g. resolution) is used across a range of temporal
and spatial scales, with configurations traceable to each other and designed to best represent the
processes which have most influence on the timescale of interest (Martin et al. 2010). The
rainfall forecast from the Met Office operational medium range (1-7 day) global model
configuration is used in this study. The Unified Model (UM) is in a process of continuous
development, taking advantage of improved understanding of atmospheric processes and steadily





increasing supercomputer power. The atmospheric component of the UM is based on non-
hydrostatic dynamics with semi-Lagrangian advection and semi-implicit time stepping. It is a
grid point model with the ability to run with a rotated pole and variable horizontal grid. A
number of sub-grid scale processes are represented, including convection (Gregory and
Rowntree 1990; Gregory and Allen 1991; Grant 2001), boundary layer turbulence (Brown et al.,
2007), radiation (Edwards and Slingo, 1996), cloud microphysics and orographic drag (Webster
et al.2003). The model is initialized using a state of the art global four-dimensional variation
(4DVAR; Rawlins et al. 2007) data assimilation technique. The year to year important changes
and upgrades during 2007–2018 in the model configuration are briefly listed in Table 1. During
2007–2018, the horizontal and vertical resolution of the global NWP configuration improved
from about 40 km and 50 levels in 2007 to about 10 km and 70 levels in 2018. A major upgrade
in the dynamical core happened in July 2014. In 2002 the ''New Dynamics'' upgrade was
implemented (Davies et al., 2005). After a decade, in July 2014, the new dynamical core named
"ENDgame" was implemented operationally at Met Office UM (Wood et al. 2014; Met
Office2014). The "ENDGame" has an advantage over its predecessor "New Dynamics" in terms
of increases in atmospheric variability. This is manifest in improved details and intensity of
large-scale storms in weather forecasts, which arises from the use of less artificial damping in the
ENDGame formulation (Met Office 2014). In addition to horizontal resolution and dynamical
core, a number of other key changes were introduced. One is the change of resolution of data
assimilation component from approximately 60 km (N216) to 40 km (N320). There is a change
to model physics which includes an increase in entrainment rate in deep convection and
improvements to several other physical parameterization schemes. The complete package is
called Global Atmosphere 6.0 (GA6) and more details are available in Walters et al. (2017).



Daily rainfall forecast up to Day-3, produced by the global operational UM used for NWP have
been evaluated over two mountainous regions of WGs and NE-states. The rainfall forecast is also
interpolated at 0.5° x 0.5° for direct comparison with the observed rainfall. The evaluation has
been restricted only over the land points to focus the model performance over land orographic
regions.
**3. Verification Approach**
Traditional verification methods such as a categorical approach are generally based on rainfall
accumulation thresholds or rainfall ranges. This approach is used by most of the operational
NWP centers to evaluate the rainfall forecast (Airey and Hulme, 1995; Wilson, 2000). When we
consider a fixed rainfall threshold or range, it is observed that the verification scores drop quite
rapidly, particularly at high threshold or range (Ashrit et al., 2015).  In general, the rainfall
distribution over different regions are inhomogeneous due to different precipitation mechanisms.
As discussed earlier about the occurrence of rainfall at different regions of India, it is very
difficult to choose the same threshold of absolute quantities to evaluate the skill of a model (in
different regions). For instance, a rainfall threshold of 5cm/day over the core monsoon Zone
(CMZ) can be considered as heavy rain (Sharma et al., 2017), which may not be the case over
the WGs and NE-states. There is a need to revisit rainfall verification based on accumulation
thresholds or ranges. To overcome this issue, Robert (2008) and Zhu et al. (2015) have used
rainfall verification based on percentiles rather than the accumulation thresholds. The purpose of
choosing the percentiles over accumulation thresholds is to remove the impact of any biases and
climatological frequencies for that region (Robert 2008; Zhu et al., 2002; Buizza et al., 2003). In
the present study, daily rainfall forecasts have been verified using the standard categorical



scores, for percentile-based thresholds. A categorical approach is based on the 2x2 contingency
table (Table 2) evaluating for different thresholds.
To evaluate the skill of the NWP forecast system, verification metrics focus on the
correspondence between the observation and forecast (Murphy, 1993). The 24-hour rainfall
exceeding $80^{th}$ and $90^{th}$ percentiles thresholds are events of interest in the present study. The
percentiles are computed over the entire period (2007-2018). Figure S1 (a) and (b) show $80^{th}$ and
$90^{th}$ percentiles rainfall in the observations. Similarly, the bottom panels, Figure S1 (c) and (d)
show $80^{th}$ and $90^{th}$ percentiles rainfall in the forecasts. These are the reference thresholds for the
evaluation. A *hit* is considered when prediction of an event matches the observation on a grid
point, while an event on a grid point predicted but not observed, we denote as a *false alarm* (b).
A *miss* (c) occurs when an event is not predicted but is actually observed. Finally, *correct*
*rejection* (d) is when an event doesn't occur and the model doesn't predict it. These four
variables are the components of the 2x2 contingency table and are displayed in Table 2. BIAS,
Probability of Detection (POD), False Alarm Ratio (FAR), Critical Success Index (CSI) and
Symmetric Extremal Dependence Index (SEDI) are some of the metrics used in this study. POD
is defined as ratio of number of correct forecasts (a) to the number of observed events (a+c)
while FAR is the ratio of number of false alarms (b) to the number of forecasts made (a + b).The
ratio of number of hits (a) to all events either forecast or observed (a + b + c) is known as CSI.
All three scores range from 0 to 1, with 1 being a perfect score in case of POD as well as CSI
and 0 for perfect FAR. The Bias Score is calculated as the ratio of the number of predicted
events (a+b) to the observed events (a+c) exceeding a given threshold (Donaldson et al., 1975).
The Bias Score ranges from 0 to infinity with a value of 1 meaning perfect forecast. The Bias
Score can help in identifying whether the forecast system has a tendency to underpredict



(BIAS<1) or overpredict (BIAS>1) events. Since the Bias Score does not provide any
information about the forecast accuracy, it is generally evaluated in conjunction with another
verification score such as Critical Success Index (CSI) or Equitable Threat Score (ETS) (Ebert et
al 2003). The detailed formulae of these metrics are displayed in Table 3 and a detailed
description can be found in Wilks (2011) and Jolliffe and Stephenson (2012). These verification
metrics have been computed for twelve monsoon seasons for rainfall exceeding $80^{th}$ (CAT-1)
and $90^{th}$ percentiles (CAT-2) over WGs and NE-states.
**4.   Results and Discussion**
**4.1. Evaluation of Forecast Rain during recent years**
The mean seasonal rainfall obtained from observations and Day-3 forecast of the UM along with
Mean Error (ME) over the Indian region for 2013, 2015 and 2018 is shown in the Figure 2. The
boxes represent the area of study used for categorical verification. We have evaluated the rainfall
for Day-1, Day-2 and Day-3 forecast but the results are shown only for Day-3 forecast for
brevity. The monsoon seasons of 2013, 2015 and 2018 are chosen to highlight the improvement
in mean rainfall forecast due to increasing the horizontal resolution and major model upgrades
discussed in section 2.2.  During JJAS of 2013 and 2015, the UM's horizontal resolution was
N512 (~25km) and N768 (~17km) respectively while the dynamical core was upgraded from
New Dynamics to ENDgame. Further, the model underwent increased horizontal resolution of
N1280 (~10km) during JJAS 2018. Although, we have evaluated the rainfall forecast for earlier
seasons during 2007-2012 also, but no significant change is found over WGs and NE-states
compared to N512 in capturing the monsoon rainfall.
As discussed, forecasting of rainfall in the tropics and Indian region, especially over the
mountainous regions of WGs and NE-states, is always a challenge. However, the NWP models



are capable of capturing the large-scale features, but again these models also fail to pick up the
fine scale features on many occasions. The UM Day-3 forecasts successfully predict the mean
high rainfall amounts along WGs with a reducing rainfall eastwards over the peninsular India,
while for rainfall over the NE-states, the model consistently shows over prediction during the
monsoon seasons of 2013, 2015 and 2018. This is quantified in terms of ME showing a wet bias
in the NE-States (extreme right panel Figure 2c, 2f, 2i). This wet bias has also been observed in
other monsoon seasons. The model shows a large wet bias in rainfall over the Indo-Gangetic
region adjoining the Himalayas during JJAS 2013, which is improved after 2013 as seen during
the monsoon seasons of 2015 and 2018. One of the possible reasons for the improvement in the
rainfall forecast over the Indo-Gangetic plains is the reduction in the UM bias for too strong
easterlies at 850 hPa (Iyengar et al., 2011) (Please see S2).
**4.2.Evaluation of Peak rainfall Forecast during recent years**
The highest rainfall of the monsoon season of 2013, 2015 and 2018 at each grid point over the
Indian regions is shown in Figure 3 from observed and model Day-3 rainfall forecasts. The top
panel shows the observed highest daily rainfall during respective seasons (Figure 3a, b and c)
while the bottom panel shows the Day-3 highest rainfall predicted by UM (Figure 3d, e and f).
During JJAS 2013, UM in Day-3 forecast fails to achieve the highest rainfall of the season
(Figure 3d) as compared to observed peak rainfall (Figure 3a) over the WGs.  This is
substantially improved in 2015 and 2018 monsoon seasons as evident in the Figure (3b, e) and
Figure (3c, f). Although, the model also shows some false alarms in this region, it consistently
retains the peak amount of rainfall in Day-3 forecasts over the NE-states. Figures 4 (a) and (b)
show the rainfall counts (>10cm/day) in observations and Day-3 forecasts over WGs and NE-
states. Over WGs, the number of counts consistently increased in Day-3 forecasts after 2011 and





it has reached closer to the observed counts in 2018 (Figure 4a).  During 2008 and 2009, the
model over-predicts the counts of rainfall exceeding 10cm/day. The number of counts has
increased over NE-states also except 2012 and 2013. The model gives an indication of over-
estimation in picking up these counts in the rest of these years (Figure 4b). The improvement in
mean rainfall (section 4.1) and highest rainfall is linked to the improved horizontal resolution in
model and data assimilation system as well as the upgrade of the dynamical core from New
Dynamics (ND) to ENDgame. Also, the revised physics package including the increase in
entrainment rate in deep convection together with improvements to several other physical
parameterization schemes lead to the improvement in the skill of UM rainfall forecast (Walters et
al. 2017. Sharma et al 2017).
**4.3. Number of counts of rainfall exceeding 80th and 90th percentiles**
As discussed before, the 80th and 90th percentile thresholds correspond to entire period 2007 to
2018. For each monsoon season, we calculate the grid point counts exceeding these thresholds as
shown in Figure 5(a) and (b). For NE, there are 475x122 grids and WG there are 403x122 grid
point counts. It is evident in Figure 5(a) that the number of grid point counts of rainfall
exceeding 80th percentiles (CAT-1) is varying from 2000 to 4000 over WG. Similarly, over NE-
states, this count varies from 1800 to 2500. Similarly, for 90th percentile threshold (CAT-2) the
counts vary from 1100 to 2100 over WG and 500 to 1500 over NE. These counts form good
sample sizes for evaluation the rainfall exceeding 80th and 90th percentiles.
**4.4. Rainfall forecast verification over WGs and NE-states using traditional verification**
**metrics**
Figures 6 and 7 display the seasonal verification scores of four metrics (BIAS, POD, FAR and
CSI) computed based on the 2x2 contingency table for two rainfall thresholds of CAT-1 and





CAT-2 over WGs and NE-states respectively. Day-1, Day-2 and Day-3 forecast have been
chosen for evaluation. It is evident from figures 6 and 7 that the prediction of orographic rainfall
during the monsoon seasons of 2007 to 2018 has been improved up to Day-3 of the forecasts
over both the regions of study for the chosen thresholds of CAT-1 and CAT-2. However, the
seasonal CSI values show a decrease with increase threshold for Day-1 to Day-3 forecasts
(Figures 6 and 7(j-l)). While analyzing the model's performance over both the mountainous
regions, CSI has a higher magnitude over WGs compared to NE-states for CAT-1 and CAT-2
thresholds.
A consistent increase (decrease) in POD (FAR) for both the rainfall thresholds of CAT-1 and
CAT-2 at all lead times clearly indicates the improvement in UM's performance in predicting
heavy (CAT-1) and very heavy rainfall (CAT-2) events over both the regions affected by
orographic rainfall (Figure 6(d-f) and 7 (d-f)). This indicates the hit rate has increased during
these monsoon years at both the rainfall thresholds of CAT-1 and CAT-2. This increase in hit
rate is due to more events being correctly predicted (Sukovich et al., 2014). Also, the reduction
in FAR indicates the improvement in POD is also due to a more accurate forecast rather than a
'spurious' increase in the number of extreme forecasts being made. This confirms that the
improvement in skill of rainfall forecast of UM during the twelve monsoon seasons is genuine
and not an artifact of more extreme rainfall forecasts being issued or the choice of verification
metrics.
The seasonal verification of frequency BIAS during JJAS 2007 to 2018 are presented in Figures
6 and 7 (a-c) over both the mountainous regions of WGs and NE-states respectively for Day-1 to
Day-3 forecast. The model accurately predicts these events of CAT-1 and CAT-2 at all lead
times during 2007 to 2018. Since the Bias Score does not provide any information about the



forecast accuracy, it is generally evaluated in conjunction with another verification score such as
Critical Success Index (CSI) which provide additional information (Ebert et al., 2003).

**4.5. Improvement in rainfall forecast : Extreme scores**

Although the traditional verification scores such as CSI discussed in previous sections depict an
improvement in the UM global operational NWP forecasting system during recent years, it tends
to zero for rare events due to its low frequency of occurrence. Consequently, the assessment of
the skill of forecasting of such heavy rainfall events is problematic because of the rarity of such
events. The verification using these categorical scores (e.g CSI, ETS, and POD) creates a
misleading impression that rare events cannot be skillfully forecast irrespective of the forecasting
system (Stephenson et al., 2008). To overcome the shortcomings of the traditional verification
metrics in predicting rare events, Ferro and Stepheson (2011) proposed a new set of verification
metrics named the Extremal Dependence Index (EDI) and Symmetric EDI (SEDI). These scores
range from -1 to 1 with 0 measuring no skill and 1 measuring the perfect score. The main
advantages in these verification metrics are their indepence of the base rate and the fact that they
do not converge to trivial values even at high rainfall events (rare events). SEDI verification
metrics for two thresholds of CAT-1 and CAT-2 during the twelve monsoon seasons are
displayed in Figure 7 (a-c) and 8 (a-c) over WGs and NE-states at all lead times. It is clear from
Figures 8 and 9 that the skill of the model has improved in predicting heavy rainfall (CAT-1) and
very heavy rainfall events (CAT-2) during the recent monsoon seasons and at all forecast lead
times. Also, the magnitude of SEDI is higher compared to traditional verification metrics (CSI)
used in the previous section. Some of the recent improvement in the UM rainfall forecast over
the mountains can be attributed to increased horizontal resolution along with improved physics
schemes and data assimilation.  A significant improvement is also evident from 2007 to 2008.



This improved skill is due to the upgrades in data assimilation system which had a full
implementation of perturbed forecast physics convection, soil moisture nudging and increase
vertical range of GPSRO data assimilation.
**Summary and Conclusions**
During the monsoon season, heavy rainfall events over the orographic regions of WGs and NE-
states of India pose a great challenge to accurate prediction using NWP models. This is mainly
due to the medium and coarser grid resolution models, which fail to accurately resolve the
orographic features and related meteorological processes. While increased grid resolution
improves heavy rainfall prediction, it often leads to forecasting excessive and unrealistic rainfall
associated with the mountains. The work reported in this paper evaluates and documents the
improved skill in the Met Office Unified Model (UM) operational global NWP rainfall forecasts
over the hilly regions of India during the monsoon seasons of 2007-2018. The changes in the
operational UM during 2007-2018 include improvements in the representation of physical
processes, improved dynamics and increased grid resolution from about 50km in 2007 to 10km
in 2018. It is rather crucial to identify and quantify the impact of improved grid resolution in
improved skill of the forecast model in predicting the heavy rains over hilly regions which are
responsible for flash floods and landslides.
Evaluation results show that UM forecasts successfully capture all the large-scale monsoon
rainfall features. The typical high rainfall amounts along the WGs and reducing rainfall amounts
eastwards over the Indian peninsula is realistic. Similarly, high rainfall amounts over the North
Eastern States with progressively reduced amounts westwards are also accurate. Evaluation
suggests some of the following significant improvements during 2007-2018.



- The large wet bias over northern India adjoining the Himalayas during 2013 is significantly reduced during JJAS 2015 and 2018.

- The highest observed rainfall amounts over WGs (>10cm/day) are completely missed in the forecasts during JJAS 2013. Following improved grid resolution and move to ENDGAME dynamical core in 2014, both of which improved the synoptic variability in the UM forecasts, the observed peak rainfall amounts (>10cm/day and also >20cm/day) are better predicted along the west coast of India during JJAS 2015 and 2018.

The verification carried out with focus on heavy (CAT-1; >80th percentile) and very heavy rainfall (CAT-2; > 90th percentile) forecasts adopts a method that takes into account the spatial variations in climatological characteristics. The main conclusions are-

- Rainfall forecast for CAT-1 has been improved by 0.18 to 0.34 (0.14 to 0.23), 0.3 to 0.5 (0.25 to 0.37) and 0.7 to 0.5 (0.75 to 0.62) in the case of CSI, POD and FAR respectively from 2007 to 2018 over WGs (NE-states) in Day-3 forecast.  Also, CSI, POD and FAR indicate an improvement from 0.1 to 0.24 (0.08 to 0.15), 0.18 to 0.38 (0.15 to 0.26) and 0.81 to 0.61 (0.84 to 0.73) for CAT-2 over WGs (NE-states). Improved skill over the WG's is higher compared to that in NE-states.

- Further, verification metrics (SEDI) for extreme and rare events have also been computed. An increase in SEDI from 0.21 to 0.55 (0.10 to 0.33) in Day-3 forecast has been noted over WGs (NE-states) in SEDI for CAT-1. The improvement in SEDI is quite impressive and is 0.19 to 0.51 (0.12 to 0.32) over WGs (NE-states) for CAT-2.

This study is based on the long record (2007-2018) of UM global model's real time rainfall forecasts over India to highlight the improved skill in heavy rainfall forecasts. More recently



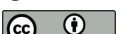

high-resolution NWP models are being used in India for operational forecasts of heavy rainfall events. Global 12km grid deterministic (NCUM) and Ensemble (NEPS; 23 members) are operational at NCMRWF. These models are also being evaluated for each season (Ashrit et al 2018) based on the 0.25 x 0.25 grid IMD-NCMRWF merged (Gauge + Satellite) rainfall analysis used in this study since higher resolution satellite-based products have biases over land and fail to capture heavy rains over land (Mitra et al., 2013). Very high resolution rainfall analysis based on all conventional rain gauges, DWR and Satellite is essential for systematic evaluation of the heavy rainfall forecasts over India.

**Code and Data Availability:**

The verification carried out in the present study uses Fortran Codes, R-Software and  verification package available in R. The observed daily rainfall data and the codes used in the study is available at ftp://ftp.ncmrwf.gov.in/pub/outgoing/kuldeep/GMED. National Center for Medium Range Weather Forecasting (NCMRWF) has an MoU with Met Office, Exeter. This **U**nified **M**odel (**UM**) forecast data can't be shared as we do receive this dataset under the mutual collaboration. However, the UM data is available for registered users on TIGGE portal (https://apps.ecmwf.int/datasets/data/tigge/levtype=sfc/type=pf/)

**Author's Contribution:**

To bring the manuscript in the final form, KS and  RA  have designed the approach of evaluation of rainfall skill over the orographic regions of India. The analysis has been carried out by KS and SK.  AKM  is the one who has developed the observed rainfall (Merged product) used in this study. ENR and SM are the principal scientists for the collaboration between NCMRWF and Met Office. KS and SK have finalized the manuscript with contributions of all the authors.





## Acknowledgements

This work and its contributor Sean Milton were supported by the Met Office Weather and
Climate Science for Service Partnership (WCSSP) India as part of the Newton Fund.

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





*Table 1. Some of the important Unified Model (UM) changes in recent years.*

| Year | UM Versions | Configurations | |
|---|---|---|---|
| | | **Resolution and Data Assimilation System** | **Dynamical Core** |
| 2007 | UM6.4 (Feb), UM6.5 (July) | N320L50 (~40 km in mid-latitudes), 12 Minute time step, 4D-VAR data assimilation | New Dynamics (ND) |
| 2008 | UM7.0 (Mar), UM7.1 (Aug) | N320L50 (~40 km in mid-latitudes), 12 Minute time step, 4D-VAR data assimilation | |
| 2009 | UM7.3 (Mar), UM7.4 (Aug) | N320L70 (~40 km in mid-latitudes), 12 minute time step, 4D-VAR data assimilation | |
| 2010 | UM7.6 (Apr), UM7.1 (Aug) | N512L70 (~25 km in mid-latitudes), 10 minute time step, 4D-VAR data assimilation | |
| 2011 | UM7.9 (Apr), UM8.0 (Aug) | N512L70 (~25 km in mid-latitudes), 10 minute time step, Hybrid 4D-VAR data assimilation | |
| 2012 | UM8.2 (Apr, PS29), UM8.2 (Sept, PS30) | N512L70 (~25 km in mid-latitudes), 10 minute time step, Hybrid 4D-VAR data assimilation | |
| 2013 | UM8.2 (Jan , PS31), UM8.2 (Apr, PS32) | N512L70 (~25 km in mid-latitudes), 10 minute time step, Hybrid data assimilation | |
| 2014 | UM8.4 (Feb, PS33) | N512L70 (~25 km in mid-latitudes), 10 minute time step, Hybrid 4D-VAR data assimilation | |
| | UM8.5 (July, PS34) | N768L70 (~17 km in mid-latitude), 7.5 minute time step, Hybrid 4D-VAR data assimilation | Even Newer Dynamics for General Atmospheric Modeling of the environment (ENDGame) |
| 2015 | UM 8.5(Feb, PS35) UM 10.1(Aug, PS36) | N768L70 (~17 km in mid-latitude), 7.5 minute time step, Hybrid 4D-VAR data assimilation | |
| 2016 | UM 10.2(Mar, PS37) UM10.4 (Nov, PS38) | N768L70 (~17 km in mid-latitude), 7.5 minute time step, | |
| 2017 | UM10.6 (Jul, PS39) | N1280L70 (~10km in Mid-latitude), 4 minute time step, Hybrid 4D-VAR data assimilation | |
| 2018 | UM10.8 (Feb, PS40) UM10.9 (Sep,PS41) | N1280L70 (~10km in Mid-latitude), 4 minute time step, Hybrid 4D-VAR data assimilation | |





*Table 2: Contingency table representing the frequencies of forecast-observation*
*pairs for which the event and non-event were forecasted and observed*

| | | Observed | | |
|---|---|---|---|---|
| | | *Yes* | *No* | *Total* |
| *Forecast* | *Yes* | Hits(a) | False alarms(b) | Forecast yes |
| | *No* | Missed(c) | Correct negatives(d) | Forecast no |
| | *Total* | Observed yes | Observed no | total |




*Table3. Categorical scores used in rainfall forecast verification in the present study*

| NAME | ACRONYMS and DEFINITIONS |
|---|---|
| BIAS | $BIAS = \dfrac{a + b}{a + c}$ |
| Probability of Detection | $POD = \dfrac{a}{a+c}$ also known as Hit Rate (H) |
| False Alarm Ratio | $FAR = \dfrac{b}{a + b}$ |
| Probability of False Detection | $POFD = \dfrac{b}{b+d}$ or known as False Alarm Rate (F) |
| Critical Success Index | $CSI = \dfrac{a}{a+b+c}$ also known as Threat Score (TS) |
| Symmetric EDI | $SEDI = \dfrac{\ln F - \ln H + \ln(1-H) - \ln(1-F)}{\ln F + \ln H + \ln(1-H) + \ln(1-F)}$ Where $H$ is hit rate and F is False Alarm Rate |




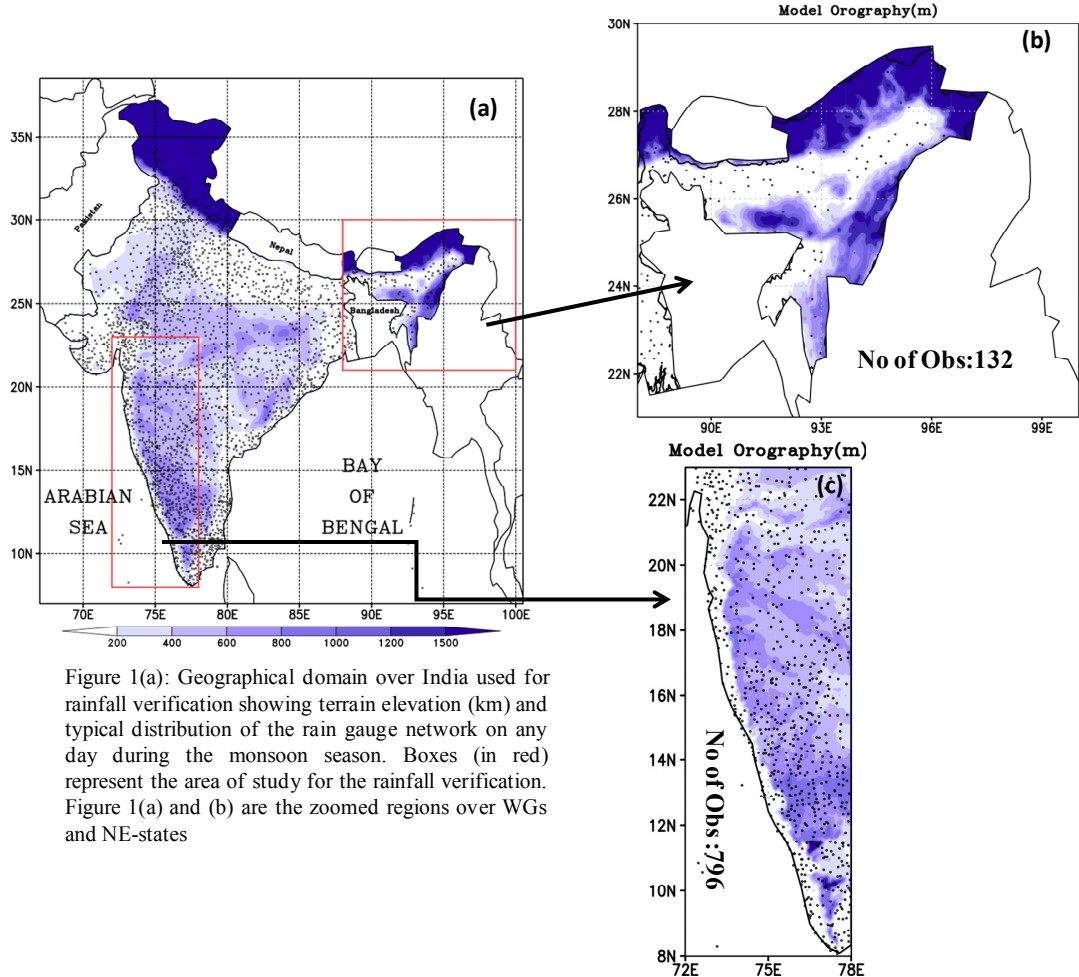

Figure 1(a): Geographical domain over India used for rainfall verification showing terrain elevation (km) and typical distribution of the rain gauge network on any day during the monsoon season. Boxes (in red) represent the area of study for the rainfall verification. Figure 1(a) and (b) are the zoomed regions over WGs and NE-states




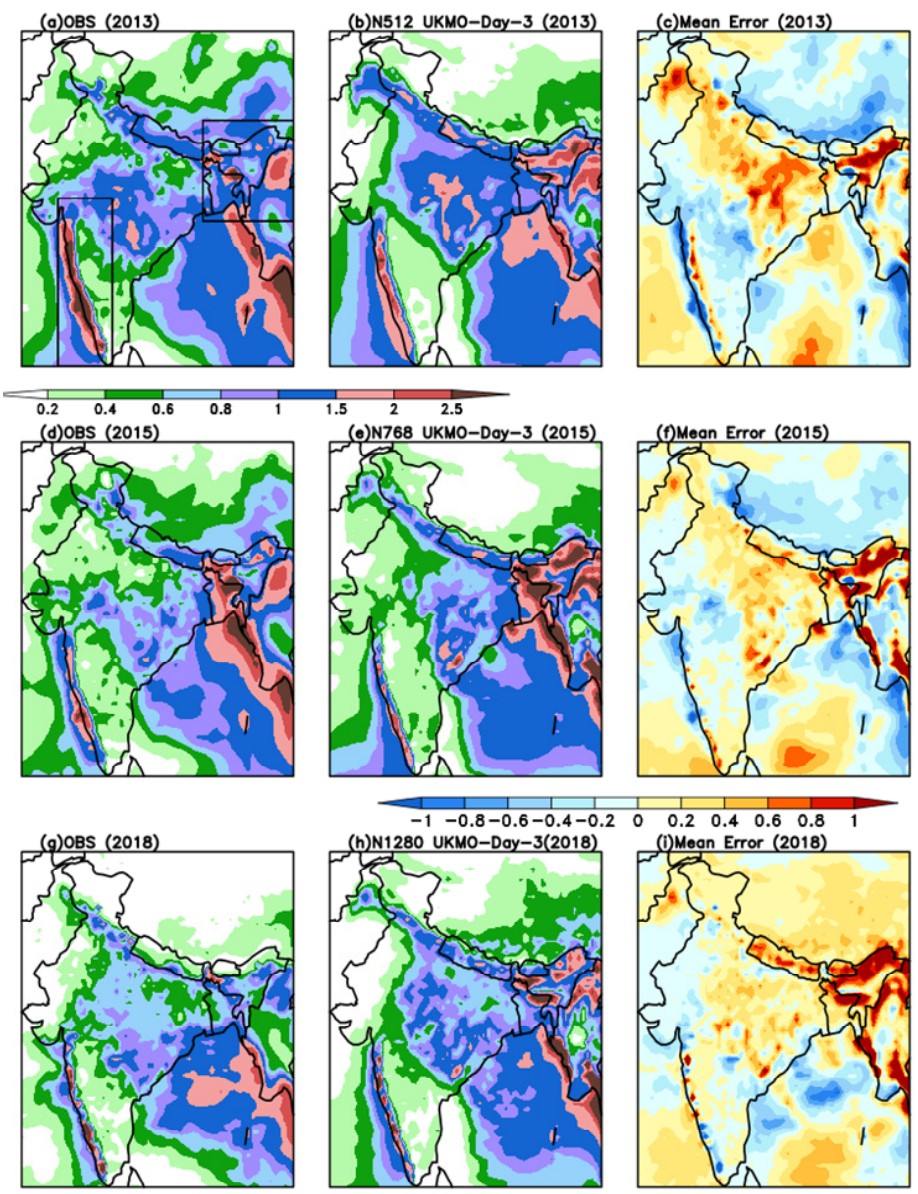

Figure 2. Observed (left panel), Day-3 Forecast  mean rainfall (middle panel) and
Mean Error (right panel)  in cm day$^{-1}$ over India during JJAS 2013, 2015 and 2018.



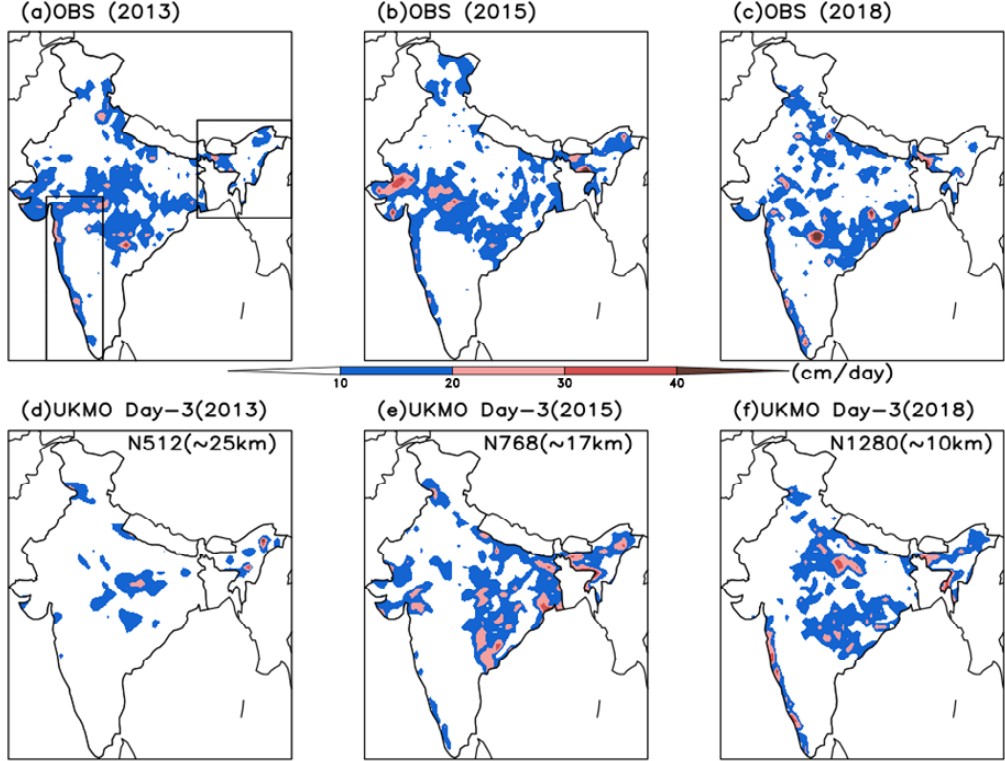


Figure 3. Observed (upper panel) and UKMO Day-3 highest rainfall Forecast (lower panel) at each grid point during JJAS 2013 , 2015 and 2018





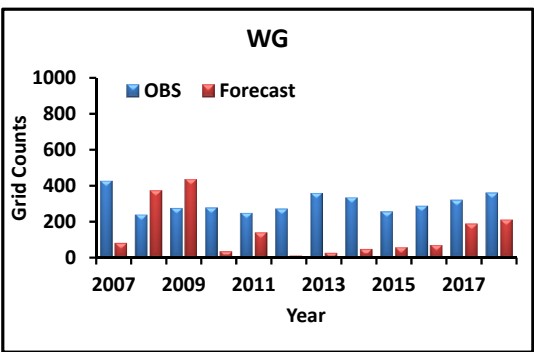
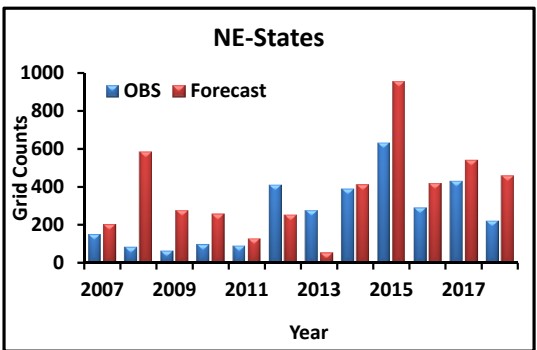


Figure 4. Number of counts in the observed and Day-3 forecast of rainfall threshold
        of 10cm/day over (a) WGs (b) NE-states






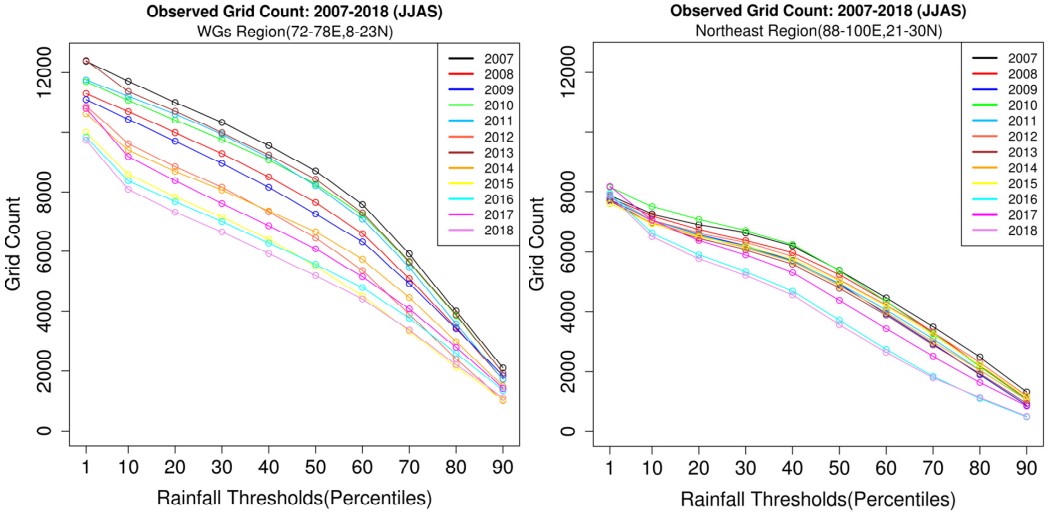


Figure 5. Observed rainfall counts over the WG and NE-states during JJAS 2007-2018.




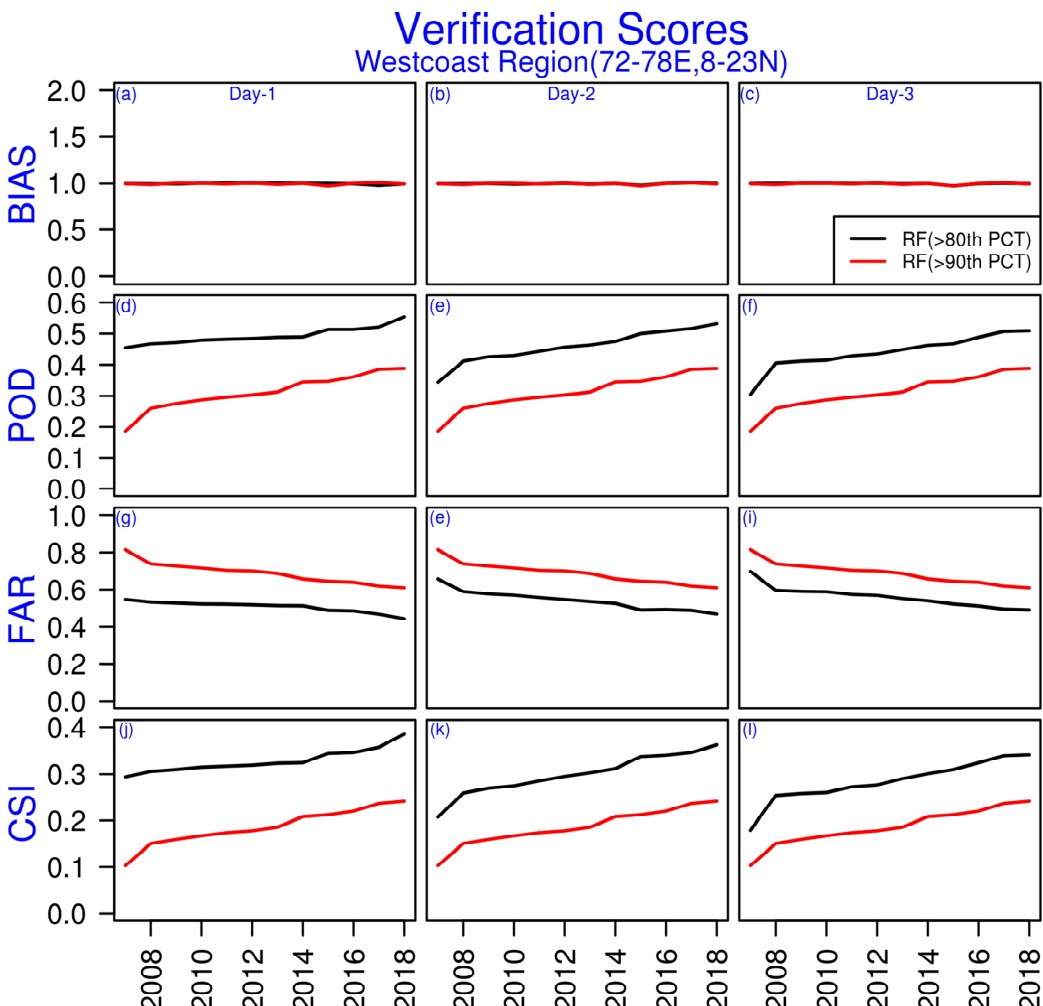


Figure 6. Bias (a-c), Probability of Detection (POD; (d-f)), False alarm Ratio
(FAR;(g-i)) and Critical success index (CSI;(j-l)) computed for Day-1 Day-2 and
Day-3 forecasts for CAT1 and CAT2 rainfall thresholds during JJAS 2007-2018
over WG .




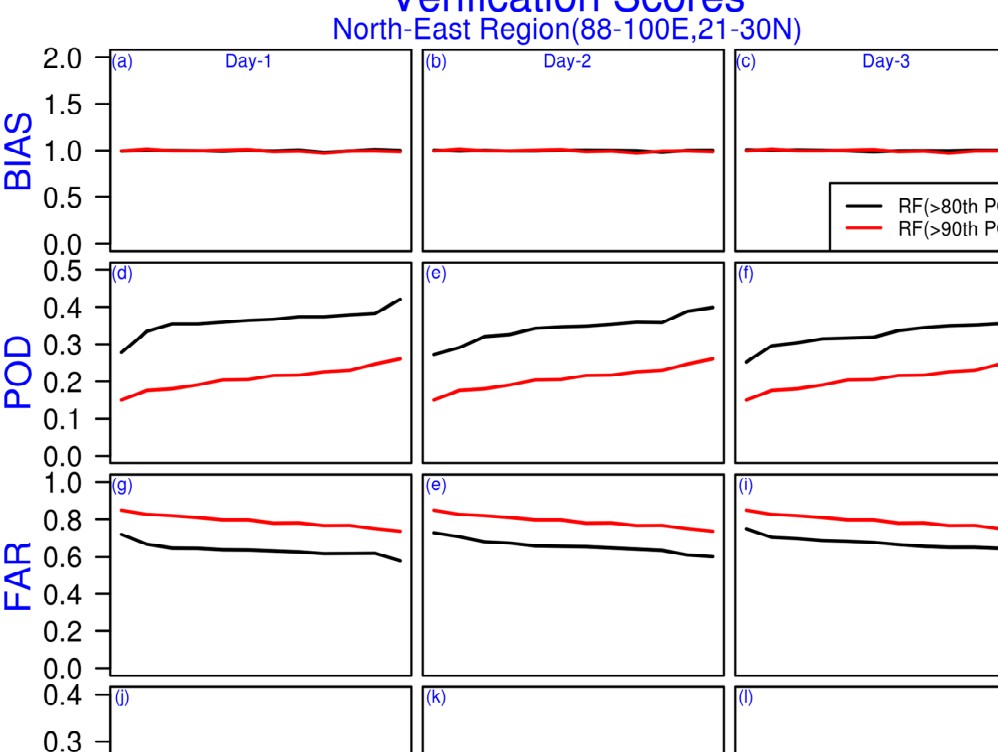


Figure 7. Bias (a-c), Probability of Detection (POD; (d-f)), False alarm Ratio
(FAR;(g-i)) and Critical success index (CSI;(j-l)) computed for Day-1 Day-2 and
Day-3 forecasts for CAT1 and CAT2 rainfall thresholds during JJAS 2007-2018
over NE-states .








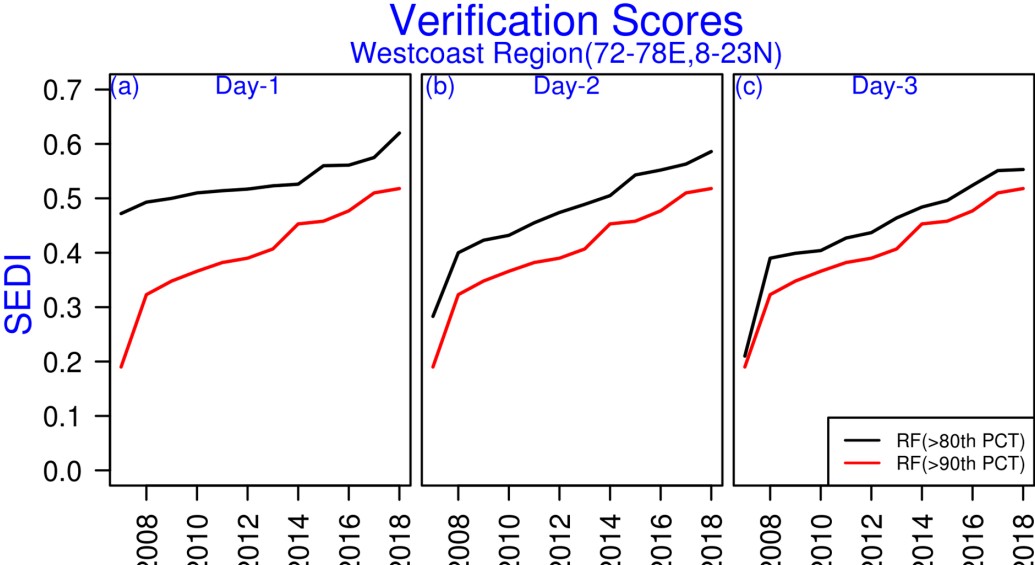


Figure 8. Symmetric extremal Dependence Index (EDI; (a-c))) computed for Day-1 Day-
      2 and Day-3 forecasts for CAT1 and CAT2 rainfall thresholds during JJAS 2007-2018
over WG region




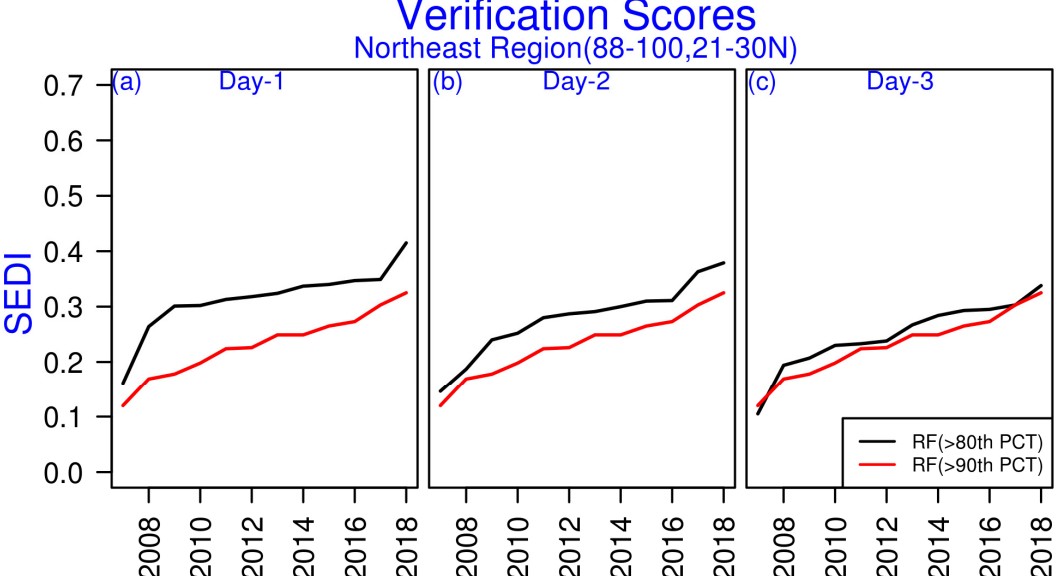


Figure 9. Symmetric extremal Dependence Index (EDI; (a-c)) computed for Day-1 Day-2 and Day-3 forecasts for CAT1 and CAT2 rainfall thresholds during JJAS 2007-2018 over NE-states