# Peer review of "Evaluation of Unified Model Rainfall Forecasts over the Western 1 Ghats and North East states of India 2 Kuldeep Sharma1, Sushant Kumar1, Raghavendra Ashrit1, Sean Milton2, Ashis K. Mitra1 and 3 Ekkattil N. Rajagopal1 4 1"

_Geoscientific Model Development, 2019_

## Referee Comment (RC1) · Anonymous Referee #1 · 22 Jun 2019

Review of "Evaluation of Unified Model Rainfall Forecasts over the Western Ghats and North East states of India" by Kuldeep Sharma, Sushant Kumar, Raghavendra Ashrit, Sean Milton, Ashis K. Mitra, and Ekkattil N. Rajagopal

Summary of manuscript:

The authors use well-established verification metrics and a newly developed satellite-gauge merged rainfall product to evaluate global Met Office Unified Model forecasts over India, with a particular focus on the Western Ghats and North-Eastern states, which are regions with steep orography. The authors use deciles (80th and 90th percentile) to identify heavy rain and very heavy rainfall events in their forecasts and obser-

vations, which are upscaled to a 0.5-degree grid. The authors find significant improvements in the forecasts over a 11-year period, during which various model improvements have taken place that could have led to such improvements.

Summary of review:

The paper is generally well written: although the grammar could be improved throughout, there are only a few places where the scientific argument is difficult to follow due to grammar. The figures are generally of good quality. However, the paper does not contain any obvious scientific advances in terms of methodology, novelty in terms of research focus, or impact of its findings on future research or model development. It does not appear to be in scope of GMD, as it comes closest to "full evaluations of previously published models", but the results presented are far removed from a "full evaluation". It is noted that a previous review indicated similarities with the Sharma et al. (2017) paper and, indeed, these similarities are significant, with only the region of interest having shifted from central India to the more mountainous terrains. Furthermore, there are flaws in the experiment design that seriously affect the impact of this paper. Taking all of this into consideration, the recommendation is to reject this manuscript for publication.

Major comments:

1. Modelling experiment design

While the authors are conscientious listing the numerous model configuration changes that have taken place during the period of study, the variety of configurations makes the intercomparison practically meaningless for scientific model development. Tracking forecast skill is a useful endeavour, but could be done by modelling centres and published through reports. In order to attribute improvements in forecast skill to a particular change in configuration – whether it is change in resolution, data assimilation practice, dynamical core, or parameterisation practice – one would expect that different configurations are run for the same period of interest. Many of the configurations studied

by the authors are no longer available, which makes findings related to those configurations practically useless for model development. One solution could be to focus on only one or two recent model configuration changes, such as a change in resolution and a change in parameterisation, and perhaps limit the period of study to the most recent monsoon seasons. Given that these forecasts are produced by another modelling centre, it is likely that these flaws in the study cannot be overcome.

**2. Focus on orography**

This has been highlighted in an earlier review and it remains a serious flaw of this study. Changes in orography – especially steep slopes as experience in the Western Ghats – will take place over scales much smaller than 0.5 degrees. How does a rainfall extreme (or decile) on the 0.5-degree scale relate to rainfall extremes at much smaller scales related to orography (valleys, catchments)? An observation-based rainfall product at such a coarse resolution will not be able to capture the detailed variations of rainfall related to orography on those scales that matter for flood forecasts, which appeared to be the motivation for this study based on the introduction. The GPM-derived IMERG rainfall product has a 0.1-degree resolution and could be more suitable for this purpose, at least for the most recent years. If the authors were to resubmit their paper, they should consider how their results are affected by the choice of observation for validation.

**3. Deciles**

This relates to the first point, but highlights the issue with experiment design. The authors consider the full period of study to derive the 80th and 90th percentile of rainfall from the observations, and do the same for the models. However, each model configuration will have its own climatological biases, leading to different deciles of rainfall. The model-derived deciles could therefore be skewed so that a configuration with a high bias will produce the model-averaged 80th percentile too frequently, while a configuration with a low bias would produce the 80th percentile very infrequently (for example). The authors demonstrate that the interannual variability leads to different frequencies

[Figure]

**GMDD**

of the multi-annual 80th decile in individual years, which complicates model evaluation further. On a similar note, it is unclear how the deciles are determined. Given that there are 475 grid point in the Western Ghats region and that there are 122 days in the JJAS period, I would expect 57950 values of daily rainfall, so 11590 counts of the 80th percentile, but the counts are around 3000. Do the authors consider the 80th percentile only conditional on there being rain measured? Looking at Figure 5, I would expect the mean counts to follow a linear relationship (each decile having approximately the same number of counts when summed over the 11-year period), but this does not appear to be the case. Why is there a change in relationship around the 60th percentile for the Western Ghats and the 40th percentile for the North Eastern states? If the authors were to resubmit the paper, ideally focusing on only two or three model configurations with changes that are relevant to the representation of rainfall over orography, they should: (1) Clearly state how the deciles are determined. (2) Determine the deciles for each model configuration and each year (the latter is already done for the observations). And (3) Report the actual values of these deciles in mm/day.

4. Bias

The authors should note that given the definition of bias = $(a+b)/(a+c)$ and given that when using deciles, e.g. the 80th percentile $(a+b)/(a+b+c+d) = 0.2$ and $(a+c)/(a+b+c+d) = 0.2$, the bias should be 1 by definition. Perhaps interannual variability and changes in model configuration (see point 3) could affect the bias, and it is peculiar that it has no discernible effect. Nevertheless, it seems uninformative to use the bias when considering deciles.

Minor comments:

Line 14-21: This is very focused on methodology and does not explain at all what is novel and what is found in this study. The authors should use the abstract to describe their results more clearly, how these results are related to model configuration choices, and how their results could lead to future model development and improvement.

[Figure]

Line 33, 39: "heavy orography rainfall" and "heavy rainfall". Please specify actual amounts.

Line 45: "This" – does this refer to "complexity" (line 42) or "observational data sets" (line 44)?

Line 56-59: Please remove the abbreviations LPS and MD, these are used only once.

Line 68: A rainfall amount for the NE is mentioned. Please include a value for the WG.

Line 74: "report improved skill" – compared to what? Also line 75 "document", what did they find?

Line 72-89: The introduction requires a brief overview of rainfall observations over India, which are clearly important for understanding orography rainfall. The first paragraph of Section 2 should be moved to the introduction. Similarly, the rationale for using quantile-based thresholds belongs in the introduction. The first paragraph of Section 3 should be moved to the Introduction.

Section 2.1: It appears that the authors do not use the same observational data consistently throughout their study. It should be made more clear to the reader that for 2007-2011, the IMD gridded data are used. Are these on a 0.5-degree grid as well? For 2012-2015, the merged data (with TRMM) are used and for 2016-2018, the merged data (with GPM) are used. How do these different products compare in terms of deciles? Are the different products available for the same period of time at any point of the period of study? This is actually quite a concern, again, for the scientific quality of this analysis.

Line 162: How are these "improvements" evaluated? These are actually changes to the parameterization schemes, but how is it determined that these are improvements to the model?

Line 164: "Daily rainfall" – what is the period of accumulation, 00-24 UTC or some other standard time?

[Figure]

Line 218: "from observations" – the authors should clarify that three different observational products are used.

Line 233: "fine scale features" – the data are on a 0.5-degree grid so it is unclear what authors mean by these features.

Line 233: "successfully predict" – how "success" determined? Subjective eyeballing of precipitation maps?

Line 235: "over prediction" – this phrase has a different meaning. Use "overestimates".

Line 242: The reference to a paper from 2011 is not appropriate to describe later model versions that are considered here (2013, 2015, 2018). The figure S2 is not clear either. How is systematic error calculated? Against analyses? How good are the analyses?

Line 245 and Figure 3: These figures would be easier to interpret if the authors reproduced two separate figures, one zoomed in to the WG and one for the NE.

Line 251: "false alarms" – this is not an appropriate phrase to use when comparing seasonally aggregated information.

Line 258-264: None of these attributions are justified without performing a systematic comparison of different model configurations for the same period of observations. See major comments.

Line 289-291: Remove "This . . . 2014)" – The authors are describing what a high POD means.

Line 297-302: Remove these lines as the Bias is not very informative when considering deciles.

Line 317: This should refer to Figure 8 and 9, not 7 and 8.

Line 320: Why is the magnitude of SEDI compared to other metrics, e.g. CSI?

Line 321-326: These claims are not supported by the findings due to the flaws in

experiment design.

Line 334: Remove "improved".

Line 338: "identify and quantify the impact of" – this is not done in this paper due to the flaw in experiment design.

Line 341-342: What are the "large-scale monsoon rainfall features"?

Line 349: Rephrase to "Following increased grid resolution. . ."

Line 350: How is the "improved synoptic variability" determined?

Line 371: Is this 0.25-degree rainfall product used in the present study?
* * *

---

## Referee Comment (RC2) · Anonymous Referee #2 · 10 Jul 2019

General Comments:

This paper presents an evaluation of general model advances which have occured in the Met Office UM since 2007, in the context of summer rainfall over India, and so the subject area is within the scope of GMD and EGU. The paper does not present any novel modelling or verification tools, nor does it report on any new model experiments. The novelty is rather in bringing together recently-developed, as well as more commonly-used, verification metrics to assess the UM in the specific context of rainfall over mountaineous regions of India, using recently-available observations.

The overall conclusion of the paper, that UM rainfall forecasts have improved steadily

over the last 12 years, for two key regions of India, is a substantial result. However, the authors provide no discussion as to how this result could have come about for reasons other than improvements in the UM. For example, it may be that different years have been easier or more difficult to forecast: to conduct a thorough evaluation, it would be necessary to run different versions of the UM for the same year and compare the forecasts. Another approach, using only the existing forecasts, might be to see whether improvements across years between which an upgrade has been made are larger than improvements across years between which an upgrade has not been made

The methods are quite clearly described, although the authors should make clear exactly which data were used to define the percentiles (e.g. was the same dataset used for both observations and forecasts, or were the observations and forecasts assigned to percentiles separately, based on their respective dataset, so that the percentiles for any given observation/forecast pair correspond to different absolute thresholds?). In terms of reproducibility of the results, the authors provide codes and data (although these could be more clearly documented, perhaps even to the extent of saying which code and data are used to produce which figure in the paper). I haven't tried to apply these, but I think that the methods and data are sufficiently clearly described in the paper that one could in principle reproduce the results from this.

The initial review brought up the issue that the 0.5 degree grid was not of sufficiently fine resolution to allow a full assessment of the forecasts in terms of orographic processes, and the authors do not seem to have addressed this. It should at least be mentioned that there are some processes that are not captured by the grid being used. On the other hand, it probably is an appropriate grid for assessing the UM (being a few times coarser than the UM grid spacing, so corresponding to the UM's "effective" resolution) and could still be appropriate for assessing larger-scale orographic processes, and the upscale effects of smaller-scale processes. Is it possible to look at the verification scores as a function of horizontal location? Then it might be possible to determine how the improvement in the forecast (as well as the quality itself of the forecasts) varies
with the steepness of the orography. Going in the opposite direction, it may be worthwhile to apply neighborhood-based verification metrics (e.g. Fractional Skill Score) to determine how the improvements vary with scale going to coarser scales.

As far as I am aware the authors give proper credit to previous work. An issue was raised in the initial review that two previous papers were very similar to this work. The main innovation of the current paper is that it looks at percentile-based thresholds, and that it looks at more mountaneous areas of India, and this is clear from the Introduction. I do wonder, however, if it would be worth comparing the percentile and absolute threshold scores directly, since the latter did not show such a pronounced improvement in the previous paper. Similarly, it may be worth applying the methods of the current paper to more regions over India.

The paper is well presented, with an appropriate title, abstract, amount of figures, supplementary material and use of mathematics. It would, however, benefit from some editing for the English: it is generally always possible to ascertain the meaning of the text, but some improvements would make it easier to read (particularly for non-native speakers of English) and therefore increase its impact. It would also be good to see Figures 2 and 3 for the other years (probably in supplementary material), and some discussion as to whether they corroborate the authors' conclusions with respect to these Figures. This is touched on in lines 227-229, where the authors say that no significant change is seen in 2007-2012, but nothing is mentioned about 2014, 2016 and 2017.

Specific Comments:

Lines 107-130: Are the satellite products not used at all for 2007-2011? This is not clear from the text. Additionally, using different observational datasets for different periods could contribute to the change in verification scores seen over different years. Is it possible to apply all three observation datasets at least to 2016-2018, to see if the scores change significantly with different datasets?

[Figure]

Lines 190-191: As mentioned in the General Comments, are these percentiles with respect to observations or forecasts? Lines 182-184 suggests that each of observations and forecasts use their own dataset to calculate the percentiles (so that the absolute thresholds are different). But this would then mean that the number of observed and forecast events is the same, so a+b=a+c and b=c. This means that BIAS=1, FAR=1-POD and CSI=POD/(2-POD). This seems to be the case from Figures 6 and 7: the quantities FAR, POD and CSI are all fairly simply related to each other and BIAS is always very close to 1. It may therefore be sufficient to only report one of these four quantities.

Line 200: Since the SEDI is a relatively new technique, it may be worth including the Ferro & Stephenson reference here aswell.

Lines 237-239: It would be useful to see a value for how the bias has changed overall for each of the two regions. This can be ascertained to some extent from the figures, but not quantitatively.

Lines 258-264: Do you have a reason why you think the improvement in mean rainfall and highest rainfall is linked to the specific upgrades you state? It is of course likely to be the case: indeed, you could probably just change "linked" to "likely to be linked" and this would be fine.

Lines 286-296: See comment above for lines 190-191. Given an improvement in POD you are guaranteed to get an improvement in FAR, if you use the percentile method where you are excluding bias effects.

Lines 317-321: Is it worth trying even higher thresholds? The more traditional methods seem to work quite well for 80% and 90%; is not the point of the SEDI to assess such extreme thresholds that insufficient counts exceeding the threshold are available to usefully apply the traditional methods?

Line 324: Does this sentence refer to the improvement from 2007-2008? This improve-

ment in the data assimilation is not listed in Table 1.

Lines 346-352: As mentioned earlier, please provide plots for other years in the Supplementary Material to support these conclusions.

Table 1: It would also be useful to list the different UM configurations used (e.g. GA6.1). Figure 4: For the NE states, the numbers of counts in more recent years are clearly higher than those in earlier years. Could this be to do with the different datasets used in different years?

Technical Corrections:

Line 112: The two numbers should be multiplied by 122 (based on text later in the manuscript).

Lines 162-163: Some earlier years of Met Office operational forecasts used an earlier version (e.g. GA3.1). I think the current version used operationally is GA6.1 but please check this.

Lines 256-257: "The number of forecast counts is too high over ...". Currently the text implies that the number of counts increases each year.

Line 317: "Figures 8 and 9".

Figure 2: Please make it clear that, for the right panels (c,f,i), you are subtracting the observations from the forecast.

Figure 4 caption: "... forecast of rainfall above a threshold of 10cm/day ..."?

Figures 6-9: Please use a range of 0 to 1 for the y-axes (except for BIAS, where the y-axis could be zoomed in further, unless you want to emphasize that this is always nearly equal to 1; see also comment for lines 190-191).

There are numerous minor grammatical errors, but these should be picked up during the copy-edit.

One of the previous papers is referenced in at least one place as "Kuldeep et al. (2017)", but appears in the reference list as "Sharma K., ...". The other previous paper is referenced as Kuldeep et al. (2019), but does not appear at all in the reference list. I found two further references that do not appear in the references list: Grant (2001) and Donaldson et al. (1975).

─────────────────────────────

---

## Author Comment (AC1) · 5 Aug 2019

Summary of review: The paper is generally well written: although the grammar could be improved throughout, there are only a few places where the scientific argument is difficult to follow due to grammar. The figures are generally of good quality. However, the paper does not contain any obvious scientific advances in terms of methodology, novelty in terms of research focus, or impact of its findings on future research or model development. It does not appear to be in scope of GMD, as it comes closest to "full evaluations of previously published models", but the results presented are far removed from a "full evaluation". It is noted that a previous review indicated similarities with the

[Figure]

Sharma et al. (2017) paper and, indeed, these similarities are significant, with only the region of interest having shifted from central India to the more mountainous terrains. Furthermore, there are flaws in the experiment design that seriously affect the impact of this paper. Taking all of this into consideration, the recommendation is to reject this manuscript for publication.

Reply to Summary of review. First of all authors are thankful for reviewing our paper and providing your feedback. As it is mentioned in the abstract, the aim of this study is to evaluate the performance of operational Unified Model (UM) rainfall forecasts for predicting heavy and very heavy rainfall events. The reviewer's remark ".…. The paper does not contain any obvious scientific advances in terms of methodology, novelty in terms of research focus, or impact of its findings on future research or model development. . .." is a bit too strong. In this study, various verification metrics (traditional scores and recently developed scores) have been used together to assess the UM rainfall forecast over India and show significant improvement in forecast skill in the specific context of rainfall over two hilly regions of India. This forms the novel application of verification scores for model evaluation.

Also the reviewer's comment "Furthermore, there are flaws in the experiment design that seriously affect the impact of this paper. Taking all of this into consideration, the recommendation is to reject this manuscript for publication."

Replies to the this comment: Perhaps this comment is not relevant and outside the scope of this study. However, all valuable suggestions and comments have been carefully studied and responses have been prepared. The manuscript is suitably modified wherever necessary.

Major comments: 1. Modelling experiment design While the authors are conscientious listing the numerous model configuration changes that have taken place during the period of study, the variety of configurations makes the intercomparison practically meaningless for scientific model development. Tracking forecast skill is a useful en-

deavour, but could be done by modelling centres and published through reports. In order to attribute improvements in forecast skill to a particular change in configuration – whether it is change in resolution, data assimilation practice, dynamical core, or parameterization practice – one would expect that different configurations are run for the same period of interest. Many of the configurations studied by the authors are no longer available, which makes findings related to those configurations practically useless for model development. One solution could be to focus on only one or two recent model configuration changes, such as a change in resolution and a change in parameterisation, and perhaps limit the period of study to the most recent monsoon seasons. Given that these forecasts are produced by another modeling centre, it is likely that these flaws in the study cannot be overcome.

Reply to Major comment 1 Modelling experiment design

The authors agree with the suggestion given by the reviewer and detailed set of experiments using a frozen version of the model needs to be done to carry out the sensitivity studies. This can be considered a separate study. The discussion presented in this paper forms a background in this paper to carry out such sensitivity studies. Thus, this paper is very relevant for planning experiments.

2. Focus on orography This has been highlighted in an earlier review and it remains a serious flaw of this study. Changes in orography – especially steep slopes as experience in the Western Ghats – will take place over scales much smaller than 0.5 degrees. How does a rainfall extreme (or decile) on the 0.5-degree scale relate to rainfall extremes at much smaller scales related to orography (valleys, catchments)? An observation-based rainfall product at such a coarse resolution will not be able to capture the detailed variations of rainfall related to orography on those scales that matter for flood forecasts, which appeared to be the motivation for this study based on the introduction. The GPM-derived IMERG rainfall product has a 0.1-degree resolution and could be more suitable for this purpose, at least for the most recent years. If the authors were to resubmit their paper, they should consider how their results are affected

by the choice of observation for validation.

Reply to Major comment 2. Focus on orography

The reviewer's comment is correct that 0.50 grid resolution is not the best to capture steep orography and associate processes. However, it can be an appropriate grid for assessing the UM (being a few times coarser than the UM grid spacing, so corresponding to the UM's "effective" resolution) for large-scale orographic processes and the upscale effects of smaller-scale processes. Also, it must be noted that this study uses the best possible rainfall product available for the entire period over Indian region. Evaluation using one or two most recent monsoon seasons will have very small sample size. GPM derived IMERG can't be used directly since it has biases in rainfall estimation (Reddy et al 2019). Also over mountains this rainfall product shows large biases. (Krishna et al 2017, doi: https://doi.org/10.1002/2017EA000285 )

3. Deciles This relates to the first point, but highlights the issue with experiment design. The authors consider the full period of study to derive the 80th and 90th percentile of rainfall from the observations, and do the same for the models. However, each model configuration will have its own climatological biases, leading to different deciles of rainfall. The model-derived deciles could therefore be skewed so that a configuration with a high bias will produce the model-averaged 80th percentile too frequently, while a configuration with a low bias would produce the 80th percentile very infrequently (for example). The authors demonstrate that the interannual variability leads to different frequencies of the multi-annual 80th decile in individual years, which complicates model evaluation further. On a similar note, it is unclear how the deciles are determined. Given that there are 475 grid point in the Western Ghats region and that there are 122 days in the JJAS period, I would expect 57950 values of daily rainfall, so 11590 counts of the 80th percentile, but the counts are around 3000. Do the authors consider the 80th percentile only conditional on there being rain measured? Looking at Figure 5, I would expect the mean counts to follow a linear relationship (each decile having approximately the same number of counts when summed over the 11-year period), but

this does not appear to be the case. Why is there a change in relationship around the 60th percentile for the Western Ghats and the 40th percentile for the North Eastern states? If the authors were to resubmit the paper, ideally focusing on only two or three model configurations with changes that are relevant to the representation of rainfall over orography, they should: (1) Clearly state how the deciles are determined. (2) Determine the deciles for each model configuration and each year (the latter is already done for the observations). And (3) Report the actual values of these deciles in mm/day.

Reply to Major comment 3 Deciles

Deciles are obtained at each grid point. Each day, there is varying number of grids. They are not summed over all 12 years period. From the figure 6, in any given year, rainfall grid counts exceeding lower deciles (10, 20, 30 . . ..) are higher than the rainfall grid counts exceeding high deciles (70, 80, 90. . .) events. This is very much consistent with figure 6.

4. Bias The authors should note that given the definition of bias = (a+b)/(a+c) and given that when using deciles, e.g. the 80th percentile (a+b)/(a+b+c+d) = 0.2 and (a+c)/(a+b+c+d) = 0.2, the bias should be 1 by definition. Perhaps interannual variability and changes in model configuration (see point 3) could affect the bias, and it is peculiar that it has no discernible effect. Nevertheless, it seems uninformative to use the bias when considering deciles.

Reply to Major comment 4. Bias

Since the number and observed forecast events is same , BIAS is close to 1. Hence, the plot is avoided. However, FAR .POD and CSI are retained.

Minor comments:

1. Line 14-21: This is very focused on methodology and does not explain at all what is novel and what is found in this study. The authors should use the abstract to describe their results more clearly, how these results are related to model configuration choices,

and how their results could lead to future model development and improvement.

Reply to Minor comment 1 We have evaluated the forecast skill for the monsoon period of 2007-2018 and also the improvement in the rainfall forecast over the period of time. The reviewer is pointing something else which we are not claiming.

2. Line 33, 39: "heavy orography rainfall" and "heavy rainfall". Please specify actual amounts.

Reply to Minor comment 2

Thank you for pointing out this as it was (heavy orography rainfall) wrongly written in the manuscript. We have modified it by removing the word "orography". Here, in the present work, rainfall exceeding 80th and 90th percentiles have been considered as heavy and very heavy rainfall.

3. Line 45: "This" – does this refer to "complexity" (line 42) or "observational data sets" (line 44)?

Reply to Minor comment 3

The text has been modified suitably and written as "Both these factors" in place of "This"

4. Line 56-59: Please remove the abbreviations LPS and MD, these are used only once. Reply to Minor comment 4 The abbreviations are removed in the text.

5. Line 68: A rainfall amount for the NE is mentioned. Please include a value for the WG.

Reply to Minor comment 5

The rainfall amount over WG has been included in the text.

6. Line 74: "report improved skill" – compared to what? Also line 75 "document", what did they find?

Reply to Minor comment 6

The text in line 74 has been modified. and "documented" the spatial verification of rainfall using Contiguous Rain Areas (CRA) method over different regions of India (add reference)

7. Line 72-89: The introduction requires a brief overview of rainfall observations over India, which are clearly important for understanding orography rainfall. The first paragraph of Section 2 should be moved to the introduction. Similarly, the rationale for using quantile-based thresholds belongs in the introduction. The first paragraph of Section 3 should be moved to the Introduction.

Reply to Minor comment 7

Thank you very much for your suggestion. The text has been modified as per the suggestions.

8. Section 2.1: It appears that the authors do not use the same observational data consistently throughout their study. It should be made more clear to the reader that for 2007-2011, the IMD gridded data are used. Are these on a 0.5-degree grid as well? For 2012-2015, the merged data (with TRMM) are used and for 2016-2018, the merged data (with GPM) are used. How do these different products compare in terms of deciles? Are the different products available for the same period of time at any point of the period of study? This is actually quite a concern, again, for the scientific quality of this analysis.

Reply to Minor comment 8

Thank you for pointing out this confusion. We wish to clarify that we have utilized only one data source (IMD-NCMRWF merged rainfall data). The text has been modified to reflect this. "The observed rainfall data is the IMD-NCMRWF merged rainfall product (0.5 x 0.5grid). This rainfall analysis is based on merging of gauges measurements with satellite based rainfall estimates (TRMM: Tropical Rainfall Measuring Mission Multi-satellite Precipitation Analysis (TMPA)-3B42 and GPM: Global Precipitation Measurement). The satellite rainfall estimates are based on TRMM during 2007-2015 and on GPM from 2016. This merged data set represents the Indian monsoon rainfall realistically and is superior to other available rainfall data sets over the Indian monsoon region (Mitra et al. 2013, Reddy et al 2019). "

9. Line 162: How are these "improvements" evaluated? These are actually changes to the parameterization schemes, but how is it determined that these are improvements to the model?

Reply to Minor comment 9

The text has been suitably modified by replacing "improvements" to "changes".

10. Line 164: "Daily rainfall" – what is the period of accumulation, 00-24 UTC or some other standard time?

Reply to Minor comment 10

The period of accumulation of rainfall is 03-03UTC to match with the rainfall observations.

11. Line 218: "from observations" – the authors should clarify that three different observational products are used.

Reply to Minor comment 11:

We have used only one rainfall product i.e. IMD-NCMRWF merge rainfall product for the entire period and text is suitably modified to reflect the same.

12. Line 233: "fine scale features" – the data are on a 0.5-degree grid so it is unclear what authors mean by these features.

Reply to Minor comment 12:

Thanks for bringing this to our notice. The discussion on "fine scale features" has been

avoided since it is not relevant.

13. Line 233: "successfully predict" – how "success" determined? Subjective eye-balling of precipitation maps?

Reply to Minor comment 13:

The discussion in line 233 and Figure 2 in the text is aimed to show the successful prediction reflected in the rainfall averaged over the season. Discussion presented in this section uses the phrase "successfully predict" is to highlight model's ability to predict heavy rains over that region. The detailed quantification of rainfall has been presented in the subsequent sections (4.4 and 4.5).

14. Line 235: "over prediction" – this phrase has a different meaning. Use "overestimates".

Reply to Minor comment 14:

The "overestimates" has been used instead of "over prediction"

15. Line 242: The reference to a paper from 2011 is not appropriate to describe later model versions that are considered here (2013, 2015, 2018). The figure S2 is not clear either. How is systematic error calculated? Against analyses? How good are the analyses?

Reply to Minor comment 15

Another reference (Iyengar et al 2014) has also been included in the text as well as in the reference list. Yes, the systematic errors are calculated against analysis.

16. Line 245 and Figure 3: These figures would be easier to interpret if the authors reproduced two separate figures, one zoomed in to the WG and one for the NE.

Reply to Minor comment 16

Two separate figures over WGs and NE-states are reproduced and it has now Figure

3(WGs) and 4 (NE-states). And also subsequent figure numbers also modified.

17. Line 251: "false alarms" – this is not an appropriate phrase to use when comparing seasonally aggregated information.

Reply to Minor comment 17

This phrase ("false alarms") has been removed from the text and modified the sentence as "Although, UM overestimates the highest rainfall over NE-states also, it consistently retains the peak rain amounts (Figure 4)"

18. Line 258-264: None of these attributions are justified without performing a systematic comparison of different model configurations for the same period of observations. See major comments.

Reply to Minor comment 18

The authors disagree with this comment. The improvement in the skill of UM rainfall may be attributed to the combined impact of increased horizontal resolution in model and data assimilation system together with revised physics package. The authors provide a background to conduct such sensitivity studies (systematic comparison of different model configurations) in the present work.

19. Line 289-291: Remove "This : : : 2014)" – The authors are describing what a high POD means.

Reply to Minor comment 19

The sentence has been removed from the text.

20. Line 297-302: Remove these lines as the Bias is not very informative when considering deciles.

Reply to Minor comment 20

With due respect to reviewer's comment, the authors disagree with this remarks. The

Frequency BIAS is assessed with CSI which is already available in the text. So, it can be retained. (Add once refenece, Beth)

21. Line 317: This should refer to Figure 8 and 9, not 7 and 8.

Reply to Minor comment 21

The figure numbers are suitably changed.

22. Line 320: Why is the magnitude of SEDI compared to other metrics, e.g. CSI?

Reply to Minor comment 22

Thank you very much for the comment. Other metrics like CSI, FAR, POD tend to have low values for verification at high thresholds which makes very difficult to evaluate and compare different models. SEDI is exclusively suited for verification of heavy and very heavy rainfall thresholds. It gives meaningful score values even for higher threshold which allows us to evaluate and compare different models.

23. Line 321-326: These claims are not supported by the findings due to the flaws in experiment design.

Reply to Minor comment 23

With due respect to reviewer's comment, the authors disagree with this remarks. As discussed in the response to major comment 1, the authors like to emphasize and clarify to the reviewer that this study does not involve any experiment. However, the present work forms the background and guidance to carry out sensitivity experiments. The detailed set of experiments using a frozen version of the model needs to be done to carry out the sensitivity studies. This can be considered a separate study.

24. Line 334: Remove "improved".

Reply to Minor comment 24

The word "improved " has been removed and sentence has been rephrased as "The

work reported in this paper evaluates and documents the skill of Met Office's operational Unified Model (UM) ( global) rainfall forecasts over the hilly regions of India during the monsoon seasons of 2007-2018."

25. Line 338: "identify and quantify the impact of" – this is not done in this paper due to the flaw in experiment design.

Reply to Minor comment 25

With due respect to reviewer's comment, the authors disagree with this remarks. This is the part of Summary and Discussion which gives importance to the work done in this study.

26. Line 341-342: What are the "large-scale monsoon rainfall features"?

Reply to Minor comment 26

Thank you for your comment on this. The meaning here is large scale mean monsoon rainfall like rainfall over monsoon trough region, high rainfall amounts over WGs and rainfall over parts of east and central India ". It has been included in the text also.

27. Line 349: Rephrase to "Following increased grid resolution: : :" 28. Line 350: How is the "improved synoptic variability" determined?

Reply to Minor comment 27 and 28:

Thank you for pointing out . The phrase "improved synoptic variability" has been removed and sentence has been modified as "The increased grid resolution and upgradation from ND to ENDGAME dynamical core in 2014 produces improvement in individual synoptic features such as troughs and tropical cyclones which are heavy rainfall systems. The peak rainfall amounts (>10cm/day) are better predicted along the west coast of India during JJAS 2015 and 2018 (Met Office 2014)."

29. Line 371: Is this 0.25-degree rainfall product used in the present study?

[Figure]

Reply to Minor comment 29:

No, 0.25x0.25 degree data has not been used in the present work. The text is modified to eliminate this confusion.

Please also note the supplement to this comment: https://www.geosci-model-dev-discuss.net/gmd-2019-65/gmd-2019-65-AC1-supplement.pdf

**Supplement:**

[Figure]

Figure S1 Observed 80 and 90th percentiles (a and c) and 80th and 90th percentiles of Day-1 rainfall forecast (b and d) during JJAS 2007-18 (cm)

[Figure]

Figure S2 . UM analysis of wind at 850hPa (left panel) and Systematic Error (right panel) in m/s over India during JJAS 2013, 2015 and 2018.

[Figure]

Figure S3. Observed (left panel), Day-3 Forecast mean rainfall (middle panel) and Mean Error (right panel) in cm day$^{-1}$ over India during JJAS 2007, 2008 and 2009.

[Figure]

Figure S4. Observed (left panel), Day-3 Forecast mean rainfall (middle panel) and Mean Error (right panel) in cm day$^{-1}$ over India during JJAS 2010, 2011 and 2012.

[Figure]

Figure S5. Observed (left panel), Day-3 Forecast mean rainfall (middle panel) and Mean Error (right panel) in cm day$^{-1}$ over India during JJAS 2014, 2016 and 2017.

[Figure]

Figure S6. Observed (upper panel) and UM Day-3 highest rainfall Forecast (lower panel) at each grid point during JJAS 2007 , 2008 and 2009 over india

[Figure]

Figure S7. Observed (upper panel) and UM Day-3 highest rainfall Forecast (lower panel) at each grid point during JJAS 2010 , 2011 and 2012 over India

[Figure]

Figure S8. Observed (upper panel) and UM Day-3 highest rainfall Forecast (lower panel) at each grid point during JJAS 2007 , 2008 and 2009 over India

---

## Author Comment (AC2) · 5 Aug 2019

General Comments: 1. This paper presents an evaluation of general model advances which have occured in the Met Office UM since 2007, in the context of summer rainfall over India, and so the subject area is within the scope of GMD and EGU. The paper does not present any novel modelling or verification tools, nor does it report on any new model experiments. The novelty is rather in bringing together recently-developed, as well as more commonly-used, verification metrics to assess the UM in the specific context of rainfall over mountainous regions of India, using recently-available observations.

[Figure]

2. The overall conclusion of the paper, that UM rainfall forecasts have improved steadily over the last 12 years, for two key regions of India, is a substantial result. However, the authors provide no discussion as to how this result could have come about for reasons other than improvements in the UM. For example, it may be that different years have been easier or more difficult to forecast: to conduct a thorough evaluation, it would be necessary to run different versions of the UM for the same year and compare the forecasts. Another approach, using only the existing forecasts, might be to see whether improvements across years between which an upgrade has been made are larger than improvements across years between which an upgrade has not been made.

Reply to General comment 1 & 2: The authors thank the reviewer's observation regarding the scope and novelty in the approach. The authors agree with the suggestion given by the reviewer and detailed set of experiments using a frozen version of the model needs to be done to carry out the sensitivity studies. This can be considered a separate study. The discussion presented in this paper forms a background to carry out such sensitivity studies. Following the suggestion (in the context of another approach), we have found that during 2011-2013, when there was no major modeling upgrades, the SEDI over NE-states (90th percentile threshold) fluctuated between 0.45 to 0.49 which crossed 0.5 and reached to 0.53 in 2015.

3. The methods are quite clearly described, although the authors should make clear exactly which data were used to define the percentiles (e.g. was the same dataset used for both observations and forecasts, or were the observations and forecasts assigned to percentiles separately, based on their respective dataset, so that the percentiles for any given observation/forecast pair correspond to different absolute thresholds?). In terms of reproducibility of the results, the authors provide codes and data (although these could be more clearly documented, perhaps even to the extent of saying which code and data are used to produce which figure in the paper). I haven't tried to apply these, but I think that the methods and data are sufficiently clearly described in the paper that one could in principle reproduce the results from this.

Reply to General Comment 3. The authors wish to clarify that the observations and forecasts use their own dataset to calculate the percentiles (different absolute threshold).

4. The initial review brought up the issue that the 0.5 degree grid was not of sufficiently fine resolution to allow a full assessment of the forecasts in terms of orographic processes, and the authors do not seem to have addressed this. It should at least be mentioned that there are some processes that are not captured by the grid being used. On the other hand, it probably is an appropriate grid for assessing the UM (being a few times coarser than the UM grid spacing, so corresponding to the UM's "effective" resolution) and could still be appropriate for assessing larger-scale orographic processes, and the upscale effects of smaller-scale processes. Is it possible to look at the verification scores as a function of horizontal location? Then it might be possible to determine how the improvement in the forecast (as well as the quality itself of the forecasts) varies with the steepness of the orography. Going in the opposite direction, it may be worthwhile to apply neighborhood-based verification metrics (e.g. Fractional Skill Score) to determine how the improvements vary with scale going to coarser scales.

Reply to General Comment 4. The authors are very much thankful about the feedback of choosing the horizontal resolution of the model. The text now included the comment on how model successfully captures larger-scale orographic processes, and the upscale effects of smaller-scale processes in the section 4.1, even when assessed at 0.5x0.5 grid resolution.

To assess the skill as a function of grid resolution, we will require some of the fuzzy verification methods (Fraction Skill Score, Scale decomposition etc). But these methods are often effective while working with very high resolution (1x1 km grid) observation and forecast hence this has not been attempted.

The exercise of computing the verification scores as a function of horizontal scale is beyond the scope of this paper.

5. As far as I am aware the authors give proper credit to previous work. An issue was raised in the initial review that two previous papers were very similar to this work. The main innovation of the current paper is that it looks at percentile-based thresholds, and that it looks at more mountainous areas of India, and this is clear from the Introduction. I do wonder, however, if it would be worth comparing the percentile and absolute threshold scores directly, since the latter did not show such a pronounced improvement in the previous paper. Similarly, it may be worth applying the methods of the current paper to more regions over India.

Reply to General Comment 5. The authors are thankful for the comments. Comparison of absolute thresholds (5 cm/day which was taken in my previous paper, Sharma et al 2017) with percentile based thresholds would be tricky since even within small domain chosen for study, absolute amounts corresponding to 90th percentile vary. Still a comparison of CSI from 2007-2018 in Day-1 forecast over NE-states is provided (Fig 1 , this no is to only for this response). CSI computed at 5cm/day threshold also shows the increasing trend from 2007 to 2018.

6. The paper is well presented, with an appropriate title, abstract, amount of figures, supplementary material and use of mathematics. It would, however, benefit from some editing for the English: it is generally always possible to ascertain the meaning of the text, but some improvements would make it easier to read (particularly for non-native speakers of English) and therefore increase its impact. It would also be good to see Figures 2 and 3 for the other years (probably in supplementary material), and some discussion as to whether they corroborate the authors' conclusions with respect to these Figures. This is touched on in lines 227-229, where the authors say that no significant change is seen in 2007-2012, but nothing is mentioned about 2014, 2016 and 2017.

Reply to General Comment 6: Thank you for your feedback about the presentation of the paper. As the reviewer suggested, we have included the supplementary figures (S3-S8). Figure S6 and S7 shows that model hardly produced the peak amounts of

rainfall during the season. Also Figure S8 shows, model shows an improvement in capturing the maximum rainfall (season's highest rainfall ) that after 2013.

Specific Comments: 1.Lines 107-130: Are the satellite products not used at all for 2007-2011? This is not clear from the text. Additionally, using different observational datasets for different periods could contribute to the change in verification scores seen over different years. Is it possible to apply all three observation datasets at least to 2016-2018, to see if the scores change significantly with different datasets?

Reply to specific Comment 1: Thank you for your feedback. The authors wish to clarify that we have utilized only one data source (IMD-NCMRWF merged rainfall data). The text has been modified to reflect this. "The observed rainfall data is the IMD-NCMRWF merged rainfall product (0.5 x 0.5grid). This rainfall analysis is based on merging of gauges measurements with satellite based rainfall estimates (TRMM: Tropical Rainfall Measuring Mission Multi-satellite Precipitation Analysis (TMPA)-3B42 and GPM: Global Precipitation Measurement). The satellite rainfall estimates are based on TRMM during 2007-2015 and on GPM from 2016. This merged data set represents the Indian monsoon rainfall realistically and is superior to other available rainfall data sets over the Indian monsoon region (Mitra et al. 2013, Reddy et al 2019). " it must be noted that this study uses the best possible rainfall product available for the entire period over Indian region. Evaluation using one or two most recent monsoon seasons will have very small sample size. GPM derived IMERG can't be used directly since it has biases in rainfall estimation (Reddy et al 2019). Also over mountains this rainfall product shows large biases. (Krishna et al 2017). Since TRMM observations are not available from 2016 . So, this exercise is also not possible at this stage.

2.Lines 190-191: As mentioned in the General Comments, are these percentiles with respect to observations or forecasts? Lines 182-184 suggests that each of observations and forecasts use their own dataset to calculate the percentiles (so that the absolute thresholds are different). But this would then mean that the number of observed and forecast events is the same, so a+b=a+c and b=c. This means that BIAS=1,
[Figure]

FAR=1- POD and CSI=POD/(2-POD). This seems to be the case from Figures 6 and 7: the quantities FAR, POD and CSI are all fairly simply related to each other and BIAS is always very close to 1. It may therefore be sufficient to only report one of these four quantities.

Reply to specific Comment 2: Yes, observations and forecasts use their own dataset to calculate the percentiles (different absolute threshold). Since the number and observed forecast events is same , BIAS close to 1, plot is avoided. However, FAR, POD and CSI are retained.

3.Line 200: Since the SEDI is a relatively new technique, it may be worth including the Ferro & Stephenson reference here as well.

Reply to specific Comment 3: The reference has been added in the text as well as in the reference list.

4.Lines 237-239: It would be useful to see a value for how the bias has changed overall for each of the two regions. This can be ascertained to some extent from the figures, but not quantitatively.

Reply to specific Comment 4: One table (Table 4) has been introduced and also text has been modified as " The mean error in 2013, 2015 and 2018 over WGs and NE-states are displayed in Table 4."

5.Lines 258-264: Do you have a reason why you think the improvement in mean rainfall and highest rainfall is linked to the specific upgrades you state? It is of course likely to be the case: indeed, you could probably just change "linked" to "likely to be linked" and this would be fine.

Reply to specific Comment 5: Thank you very much for this comment. The text has been modified as suggested.

6. Lines 286-296: See comment above for lines 190-191. Given an improvement in POD you are guaranteed to get an improvement in FAR, if you use the percentile

method where you are excluding bias effects.

Reply to specific Comment 6: Thank you for your observation. When we consider percentile based threshold, BIAS becomes 1. Hence, BIAS is implicitly removed.

7.Lines 317-321: Is it worth trying even higher thresholds? The more traditional methods seem to work quite well for 80% and 90%; is not the point of the SEDI to assess such extreme thresholds that insufficient counts exceeding the threshold are available to usefully apply the traditional methods?

Reply to specific Comment 7: Thank you for your feedback on this. As the reviewer has pointed out the use of SEDI for higher thresholds, we have computed SEDI for 95th and 99th percentile threshold also. We found the same increasing trend. The figures show the SEDI for Day-1, Day-2 and Day-3 forecasts for 95th and 99th percentiles over WGs (Fig 2, this no is to only for this response) and NE-states (Fig 3, this no is to only for this response).

8.Line 324: Does this sentence refer to the improvement from 2007-2008? This improvement in the data assimilation is not listed in Table 1.

Reply to specific Comment 8: Thanks for pointing this out. We have now included it in the Table 1. The revised version of Table 1 has now new Table 1.

9.Lines 346-352: As mentioned earlier, please provide plots for other years in the Supplementary Material to support these conclusions.

Reply to specific Comment 9: The plots for earlier monsoon seasons from 2007-2012 has been included in the supplementary material which also confirms the consistent wet bias over Indo-Gangetic plains.

10.Table 1: It would also be useful to list the different UM configurations used (e.g. GA6.1). Figure 4: For the NE states, the numbers of counts in more recent years are clearly higher than those in earlier years. Could this be to do with the different datasets used in different years?

[Figure]

Reply to specific Comment 10: The different UM configurations has been added in Table 1. As clarified earlier, there are no different data sets. It is the merged (satellite +gauge) rainfall product. These higher counts in some years may be due to the interannual variability.

Technical Corrections:

1.Line 112: The two numbers should be multiplied by 122 (based on text later in the manuscript). Reply to Technical Correction 1:

As indicated by the reviewer, The numbers now have been multiplied by 122. Also, The text has been modified accordingly.

2.Lines 162-163: Some earlier years of Met Office operational forecasts used an earlier version (e.g. GA3.1). I think the current version used operationally is GA6.1 but please check this. Reply to Technical Correction 2:

Thank you for pointing out. Yes, It is GA6.1 and it has been modified in the text also.

3.Lines 256-257: "The number of forecast counts is too high over ...". Currently the text implies that the number of counts increases each year.

Reply to Technical Correction 3: The text has been modified as suggested.

4.Line 317: "Figures 8 and 9".

Reply to Technical Correction 4: Thank you for pointing out. The text has been modified

5.Figure 2: Please make it clear that, for the right panels (c,f,i), you are subtracting the observations from the forecast.

Reply to Technical Correction 5: Yes, the mean error is computed by subtracting observation from the forecasts. The formula has been added in the Table 3.

6.Figure 4 caption: "... forecast of rainfall above a threshold of 10cm/day ..."?

Reply to Technical Correction 6: This figure now has changed to figure 5 and the

caption has been modified as suggested.

7.Figures 6-9: Please use a range of 0 to 1 for the y-axes (except for BIAS, where the y-axis could be zoomed in further, unless you want to emphasize that this is always nearly equal to 1; see also comment for lines 190-191).

Reply to Technical Correction 7: As suggested by the reviewer, we have changed the range of yaxis (0 to 1). The plot of BIAS has been removed in context to reply of specific comment 2.

8.One of the previous papers is referenced in at least one place as "Kuldeep et al. (2017)", but appears in the reference list as "Sharma K., ...". The other previous paper is referenced as Kuldeep et al. (2019), but does not appear at all in the reference list. I found two further references that do not appear in the references list: Grant (2001) and Donaldson et al. (1975).

Reply to Technical Correction 8: Thank you for pointing out. All the references are correctly included in the text and reference list.

Please also note the supplement to this comment:
https://www.geosci-model-dev-discuss.net/gmd-2019-65/gmd-2019-65-AC2-supplement.pdf

―――――――――――――――――――――――

[Figure]

[Figure]

**Fig. 1.**

[Figure]

**Fig. 2.**

[Figure]

**Fig. 3.**

**Supplement:**

[Figure]

Figure S1 Observed 80 and 90th percentiles (a and c) and 80th and 90th percentiles of Day-1 rainfall forecast (b and d) during JJAS 2007-18 (cm)

[Figure]

Figure S2 . UM analysis of wind at 850hPa (left panel) and Systematic Error (right panel) in m/s over India during JJAS 2013, 2015 and 2018.

[Figure]

Figure S3. Observed (left panel), Day-3 Forecast mean rainfall (middle panel) and Mean Error (right panel) in cm day$^{-1}$ over India during JJAS 2007, 2008 and 2009.

[Figure]

Figure S4. Observed (left panel), Day-3 Forecast mean rainfall (middle panel) and Mean Error (right panel) in cm day$^{-1}$ over India during JJAS 2010, 2011 and 2012.

[Figure]

Figure S5. Observed (left panel), Day-3 Forecast mean rainfall (middle panel) and Mean Error (right panel) in cm day$^{-1}$ over India during JJAS 2014, 2016 and 2017.

[Figure]

Figure S6. Observed (upper panel) and UM Day-3 highest rainfall Forecast (lower panel) at each grid point during JJAS 2007 , 2008 and 2009 over india

[Figure]

Figure S7. Observed (upper panel) and UM Day-3 highest rainfall Forecast (lower panel) at each grid point during JJAS 2010 , 2011 and 2012 over India

[Figure]

Figure S8. Observed (upper panel) and UM Day-3 highest rainfall Forecast (lower panel) at each grid point during JJAS 2007 , 2008 and 2009 over India